# Recent Advances in the Development of Lipid-, Metal-, Carbon-, and Polymer-Based Nanomaterials for Antibacterial Applications

**DOI:** 10.3390/nano12213855

**Published:** 2022-11-01

**Authors:** Ruohua Ren, Chiaxin Lim, Shiqi Li, Yajun Wang, Jiangning Song, Tsung-Wu Lin, Benjamin W. Muir, Hsien-Yi Hsu, Hsin-Hui Shen

**Affiliations:** 1Department of Materials Science and Engineering, Faculty of Engineering, Monash University, Clayton, VIC 3800, Australia; 2Biomedicine Discovery Institute, Department of Biochemistry and Molecular Biology, Monash University, Clayton, VIC 3800, Australia; 3College of Chemistry & Materials Engineering, Wenzhou University, Wenzhou 325035, China; 4Department of Chemistry, Tunghai University, No.1727, Sec.4, Taiwan Boulevard, Xitun District, Taichung 40704, Taiwan; 5CSIRO, Manufacturing, Clayton, VIC 3169, Australia; 6School of Energy and Environment, Department of Materials Science and Engineering, City University of Hong Kong, Kowloon Tong, Hong Kong 518057, China

**Keywords:** nanomaterials, multidrug-resistant bacteria, antimicrobial, drug delivery systems, nanoparticles

## Abstract

Infections caused by multidrug-resistant (MDR) bacteria are becoming a serious threat to public health worldwide. With an ever-reducing pipeline of last-resort drugs further complicating the current dire situation arising due to antibiotic resistance, there has never been a greater urgency to attempt to discover potential new antibiotics. The use of nanotechnology, encompassing a broad range of organic and inorganic nanomaterials, offers promising solutions. Organic nanomaterials, including lipid-, polymer-, and carbon-based nanomaterials, have inherent antibacterial activity or can act as nanocarriers in delivering antibacterial agents. Nanocarriers, owing to the protection and enhanced bioavailability of the encapsulated drugs, have the ability to enable an increased concentration of a drug to be delivered to an infected site and reduce the associated toxicity elsewhere. On the other hand, inorganic metal-based nanomaterials exhibit multivalent antibacterial mechanisms that combat MDR bacteria effectively and reduce the occurrence of bacterial resistance. These nanomaterials have great potential for the prevention and treatment of MDR bacterial infection. Recent advances in the field of nanotechnology are enabling researchers to utilize nanomaterial building blocks in intriguing ways to create multi-functional nanocomposite materials. These nanocomposite materials, formed by lipid-, polymer-, carbon-, and metal-based nanomaterial building blocks, have opened a new avenue for researchers due to the unprecedented physiochemical properties and enhanced antibacterial activities being observed when compared to their mono-constituent parts. This review covers the latest advances of nanotechnologies used in the design and development of nano- and nanocomposite materials to fight MDR bacteria with different purposes. Our aim is to discuss and summarize these recently established nanomaterials and the respective nanocomposites, their current application, and challenges for use in applications treating MDR bacteria. In addition, we discuss the prospects for antimicrobial nanomaterials and look forward to further develop these materials, emphasizing their potential for clinical translation.

## 1. Introduction

Antibiotics have been the primary treatment choice for use on bacterial infections due to their cost efficiency and powerful and fast-acting outcomes. However, bacteria possess the intrinsic ability to evolve rapidly through mutations in developing resistance to these treatments. In addition, bacteria can transfer drug-resistant genes among their community through horizontal gene transfer, resulting in the emergence of multidrug-resistant (MDR) bacteria, which are widely known as superbugs as defined by the medical and research communities [1]. Since bacterial resistance emerges and spreads via the acquisition of genetic material from resistant bacterial cells, the evolution of antibiotic resistance is unstoppable [2]. Recent projections indicate that a post-antibiotic era is approaching, and this will result in approximately 10 million annual deaths by 2050 from MDR bacterial infections [3]. Studies have shown that infections caused by multidrug-resistant bacteria cause more harm and higher patient mortality than infections caused by susceptible strains of the same species [4]. A continual increase in the numbers of infections resulting from such resistant strains poses a serious threat globally [5]. 

The antibiotic resistance crisis is further complicated by a lack of new antibacterial agents to act as last-line defenders for the treatment of MDR bacterial infections. For instance, the World Health Organization (WHO) has identified 80 antibacterial agents that are under clinical development to treat top-priority MDR bacteria up to November 2021, but most of these are modifications of current antibiotics and will act merely as short-term solutions [6]. Only seven of these antibacterial agents are novel chemical entities that will contribute to expanding the current antibiotic pipeline [6]. Due to economic and regulatory hurdles, the biopharmaceutical industry has largely withdrawn from developing new antibiotics, further exacerbating the situation [7]. This has triggered initiatives worldwide to discover and exploit novel antibacterial agents in order to prevent these infections from happening and to overcome the current challenges faced from MDR infections [8]. Promising solutions for the prevention and treatment of MDR bacterial infections are under investigation, such as nanotechnology and biomaterials [9].

Nanotechnology serves as an alternative promising solution for the prevention and treatment of MDR bacterial infection. Nanotechnology plays an important role in this area by covering a broad range of nanostructured materials that possess inherent antibacterial activity. Nanomaterials also show significant potential for delivering drugs to specific targeted sites in vivo [10]. Nanomaterials have at least one dimension in the nano range (1–100 nm) that convey particular and variable physiochemical properties from their bulk constituents [11]. The nanosized scale of these nanomaterials can result in multivalent interactions with bacteria, including electrostatic attractions, hydrophobic and receptor–ligand interactions, and van der Waals (hydrophobic) forces [12]. This offers particular advantages compared to small molecule antibiotics that typically result in a single mode of interaction. The ease of functionalization and engineering of nanomaterials confers them with additional advantages for mechanistically overcoming bacterial resistance [13].

Nanomaterials can be broadly classified into organic nanomaterials and inorganic nanomaterials [14]. Recent advances of nanotechnology have brought novel understandings in using nanosized building blocks to design and create new nanocomposites or nanohybrid materials with unprecedented physical properties and enhanced antibacterial activity [15]. A variety of nanomaterials can be combined to develop new nanocomposite materials, with the most-established examples being depicted in the section below. In this review, we illustrate that each category of these antibacterial nanomaterials has its own distinctive characteristics and properties which are being applied to various antibacterial applications. We present recent advances in developing the use of these nanomaterials in combating MDR bacterial infections. However, the use of nanocomposites is still at an early stage and more research and investment is needed towards these efforts before we start seeing outcomes from their clinical translation. Based on these, this review summarizes previous research progress on nanotechnology in antibacterial aspects which focuses on the last 5 years, including a detailed summary and comparison of the most promising and interesting nanomaterials (Table 1). The aim is to inspire future research ideas in this field by identifying gaps or inconsistencies in the body of knowledge.

## 2. Organic Nanomaterials

Organic nanomaterials usually comprise carbon and hydrogen atoms that form, most simply, hydrocarbon-based molecules. Organic nanomaterials can be designed to those that may self-assemble into nanostructures with different dimensionalities or desired characteristics by utilizing the weak intermolecular interactions of organic molecular structures [28]. Organic nanomaterials can be classified into lipid-based, polymer-based, and carbon-based nanomaterials, and these nanomaterials can be designed to act as nanocarriers or antibacterial agents in antibacterial applications.

### 2.1. Lipid-Based Nanomaterials

A variety of lipid candidates, including free fatty acids, phospholipids, glycolipids, sphingolipids, fatty alcohols, glycerol esters, and waxes, can be utilized to nanoformulate into different classes of lipid-based nanoparticles including liposomes, emulsions, solid-lipid nanoparticles, and nanostructured lipid carriers [29]. A detailed review of these aforementioned nanoparticles has been described elsewhere [30,31,32] and will not be discussed here. Lipid-based nanoparticles are the most-established nanocarriers investigated for the delivery of a variety of pharmaceutical agents with different solubilities and pharmacokinetic behaviors. In addition to the role of nanocarrier, lipid-based nanoparticles have an emerging role as antibacterial agents against MDR bacteria. In short, these can be classified into lipidic nanocarriers and lipidic nanoparticles. Lipidic nanocarriers contain and deliver antibacterial agents including antibiotics and antimicrobial peptides, whilst lipidic nanoparticles themselves display inherent antibacterial properties.

#### 2.1.1. Lipidic Nanocarriers as Delivery Vehicles for Antimicrobial Agents

Lipidic nanocarriers are the most-established nanocarriers utilized for delivery of a variety of pharmaceutical agents with different solubilities. Lipidic nanocarriers are composed of colloidal dispersions of physiological or physiological-related lipids (natural or synthetic lipids that have the similar chemical structure to physiological lipids) in aqueous solution. Generally, these dispersions are stabilized by an emulsifier or surfactant which intercalates on the lipid nanoparticle’s surfaces. This provides the nanoparticle stability by conferring steric stabilization in between the nanoparticles and reducing the interfacial energy between the lipidic nanoparticles and the aqueous phase [33]. In brief, lipidic nanocarriers such as liposomes [16,34,35,36,37], micelles [38,39,40], nanocapsules [41,42,43,44], emulsions [45,46], and solid lipid nanoparticles [29,47] have several advantages for delivering antimicrobial agents [48]. Lipidic nanocarriers can exhibit good biocompatibility and non-immunogenic properties due to the analogous behavior of the physiological or physiological-related lipids to biological membranes as seen in the new SARS-CoV-2 lipid-based mRNA vaccines. The encapsulation of drugs enhances their bioavailability, increases the feasibility for various routes of administration, reduces associated drug toxicity, and protects the drugs from metabolic degradation. Furthermore, drug encapsulation into lipid-based nanoparticles also improves the pharmacokinetic and pharmacodynamic profiles, which lowers the required dosages and improves the therapeutic index. Lastly, surface modifications of lipid-based nanoparticles can be achieved for various purposes, such as targeted therapies, improved cellular uptake, and increased circulation times and half-lives. 

The use of these lipidic nanocarriers for delivery can have some limitations, including occasional poor colloidal or thermodynamic/kinetic stability for long-term storage, high membrane permeability that accounts for drug leakage, and low entrapment efficiency for certain hydrophobic drugs [17]. This often leads to costly and restricted preparation conditions that allows reconstitution of lipidic nanocarriers in solution prior to administration [49]. One of the potential solutions is to combine lipid-based nanoparticles with polymeric nanomaterials, forming lipid–polymer hybrid nanoparticles for delivery purposes, which will be described in Section 4.5. Despite being the most widely explored nanoparticulate delivery system for various pharmaceutical products, the role of lipidic nanocarriers in antibacterial application is limited. Currently, only one liposomal formulation (amikacin liposome inhalation suspension, Arikayce) is approved by the Food and Drug Administration (FDA) (ClinicalTrials.gov Identifier: NCT01316276) for the treatment of mycobacterial lung infection. In addition, there are a few liposomal nanoformulations delivering antibacterial agents that are undergoing clinical trials, with the details shown in Table 2.

As compared with liposomes, other lipidic nanocarriers are still in the early stages of development [50,51,52,53]. Recently, non-lamellar lyotropic liquid crystalline nanoparticles including cubosomes and hexosomes have emerged to be the next generation of smart lipidic nanoparticles [54,55,56,57] for antimicrobial therapeutics. The antibiotic potential of cubosomes with a series of magnetite (Fe_3_O_4_), copper oxide (Cu_2_O), and silver (Ag) nanocrystals were developed by Meikle et al. [58]. The results showed that Ag nanocrystal-embedded cubosomes displayed exhibitory activity against both Gram-positive and Gram-negative bacteria, with observed minimum inhibitory concentration values ranging from 15.6–250 μg/mL. Recent studies have shown that polymyxin-loaded cubosomes can enhance antibacterial potency against Gram-negative bacteria, including polymyxin-resistant strains, and enable an alternative strategy for treating pathogens by combining cubosomes with polymyxins as a combination therapy [57]. To overcome the difficulty of using antimicrobial peptides in antibiotic therapies due to their lack of specificity and their susceptibility to in vivo proteolysis, Boge et al. used cubosomes to topically deliver the antimicrobial peptides, LL-37, to inhibit *S. aureus*. They found that the pre-loading preparation where incorporation of LL-37 into liquid crystal gels followed by dispersion into nanoparticles was most effective in killing *S. aureus* [55]. Additional studies have been reported investigating the use of cubosomes as drug delivery vehicles for LL-37. It was observed that the cubosomes successfully protected LL-37 from proteolytic degradation with significantly enhanced bactericidal effects against Gram-negative strains [59]. Meikle et al. explored the potential of cubosomes as delivery vehicles for six different antimicrobial peptides, including gramicidin A, alamethicin, melittin, indolicidin, pexiganan, and cecropin A [60], wherein it was observed that by adding physiological concentrations of anionic lipids or NaCl to screen the electrostatic charge of peptides, the antimicrobial peptides loading efficiency of the cubosomes was significantly improved, and encapsulation in the cubosome carriers was shown to enhance the antimicrobial activity of certain formulations [60]. Notably, there are fundamental differences in the mechanism of cubosomes uptake between Gram-positive and Gram-negative bacteria. For Gram-positive bacteria, the cubosomes adhere to the exopeptidoglycan layer and slowly internalize into the bacteria, while for Gram-negative bacteria, the interaction occurs in two stages: the cubosomes fuse with the outer lipid membrane and then pass through the inner wall via diffusion [61].

#### 2.1.2. Lipidic Nanoparticles with Inherent Antibacterial Activities

Antimicrobial lipids composed of a carboxylic acid group and a saturated or unsaturated carbon chain (Figure 1) can act as surfactants via a membrane lytic mechanism [62]. Antimicrobial lipids possess broad-spectrum antibacterial activities and serve as new and attractive candidates to fight the antibiotic resistance crisis. However, some technical challenges impede the in vivo activity of antimicrobial lipids in bulk form. These include poor aqueous solubility and weaker in vivo bactericidal activity due to in vivo oxidation, esterification, and lipid–protein complexation [10,63,64]. This can be overcome by developing lipid nanoparticle technologies to encapsulate antimicrobial lipids and convert them into different nanoformulations with inherent antimicrobial activities. The resulting antimicrobial lipidic nanoparticles using nanocarriers have excellent water solubility, can provide high concentrations of antibacterial lipids, and protect antibacterial lipids from degradation, which highlights the great potential for improving the therapeutic ability of antibacterial lipids [10,62]. Several reviews have been published elsewhere to understand the composition, mechanism, and characterization of this class of lipidic nanoparticles [65,66,67,68,69].

Liposomal formulations are the most-studied candidate so far for the emerging role as antimicrobial agents which are spherical closed lipid bilayers that can self-assemble in aqueous solutions and have a water core [16,70]. Antimicrobial lipids such as lauric acid and oleic acid can be incorporated to form antimicrobial liposomal formulations against *Propionibacterium acnes* and methicillin-resistant *S. aureus* (MRSA), respectively [34,71]. Among these different fatty acids, liposomal linolenic acids have received considerable attention by exhibiting particularly high levels of inhibitory activity [70]. Liposomal linolenic acid (LLA, Figure 2) that comprised liposomal nanoparticles made from linolenic acid, phospholipids, and cholesterols eradicated *Helicobacter pylori* clinical isolates including metronidazole-resistant *H. pylori* [72]. Furthermore, the bacteria did not appear to develop resistance to LLA at the sub-bactericidal concentrations used when compared with metronidazole and free linolenic acid. The fusion between the LLA and bacterial membrane, which directly inserts the linolenic acid into the bacterial membranes for subsequent membrane lysis, is suggested to be the bactericidal mechanism [72]. The in vivo efficacy of LLA in treating *H. pylori* infection was further investigated [73]. LLA penetrated into the mucus layer of a murine stomach, which led to reduced bacterial load and proinflammatory cytokines. In addition, a significant portion of LLA remained in the stomach at 24 h post-treatment, showing the long-last effects of LLA. Lastly, the in vivo toxicity showed no significant increase in gastric epithelial apoptosis and no changes of the murine gastric tissue under histological analysis, indicating the excellent biocompatibility of LLA in the stomach of control mice [73].

In addition to antimicrobial liposomal formulations, other antimicrobial lipid-based nanoparticle systems, including emulsions and solid lipid nanoparticles, have shown promising antibacterial effects against MRSA and *Pseudomonas aeruginosa*, respectively [47,74]. Sadiq et al. encapsulated nisin in monolaurin nano-emulsions and demonstrated their ability of effectively inhibiting *S. aureus* in vitro [46]. Studies have found that solid lipid nanoparticles loaded with retinoic acid and lauric acid inhibited the growth of *Staphylococcus epidermidis*, *P. acnes,* and *S. aureus* [75]. A second generation of lipid nanoparticles that can improve the loading capacity and inhibit the excretion of bioactive compounds, called nanostructured lipid carriers, was recently developed from a mixture of solid lipids and liquid lipids [76,77,78,79,80]. Compared to the crystalline lipid core of solid lipid nanoparticles, the structural imperfections of nanostructured lipid carriers with less ordered crystalline arrangement can further improve the loading capacity and prevent the drug leakage for better antibacterial activity. Previous research comparing the antibacterial activity of docosahexaenoic acid (DHA) coated by nanostructured lipid carriers and DHA itself has found the incorporation of DHA into the nanostructured lipid carriers greatly enhanced bactericidal effect against *H. pylori* [81]. However, studies of emulsions, solid lipid nanoparticles, nanostructured lipid carriers, etc., as antimicrobial lipid-based nanoparticle systems are still in the early phases compared with the simplest form of liposomes.

### 2.2. Biodegradable Polymeric Nanomaterials 

Biodegradable polymeric nanosystems can be classified into polymeric nanoparticles for the purposes of a delivery nanocarrier and antimicrobial polymers. The tailored design of polymeric chains confers versatile functions to the biodegradable polymeric nanomaterials including antibacterial activity, enhancing stability, biocompatibility, long circulation, and specific bacterial recognition of the polymeric nanomaterials [82]. Antimicrobial cationic polymers are the most-studied organic nanomaterials that have already entered clinical trials and hold great promise in replacing some antibiotics [83]. Biodegradable polymeric nanoparticles also offer an attractive delivery system which can improve the safety and efficacy of other ingredients by modulating the rate, timing, and location of release compared to lipid-based nanoparticles [84]. Furthermore, the functional groups on the polymer chain serve as a promising matrix to interact with other nanomaterials, forming polymer-based nanocomposites [85]. This paves the way for researchers to synthesize different polymer-based nanocomposites with improved or novel properties, which will be discussed further in Section 4 below. 

#### 2.2.1. Polymeric Nanoparticles as Delivery Nanocarriers

For the use of biodegradable polymeric nanoparticles with encapsulated antibacterial agents, they can be classified into four distinct classes including polymeric micelles, vesicles, nanocapsules, and nanospheres, depending on the polymer composition and the final structure of the polymeric nanosystems. A detailed review of the aforementioned nanoparticles has been described elsewhere [85,86,87,88]. Currently, over 80 clinical trials are underway or have been completed using polymeric nanoparticles in cancer therapy, highlighting the potential utility of polymeric nanoparticles in drug delivery [89].

Polymeric nanoparticles and lipid-based nanoparticles share similar advantages as drug delivery vehicles, but polymeric nanoparticles have some perceived advantages over lipid-based nanoparticles. This includes higher structural integrity and stability under biological and storage conditions, and controlled release capabilities conferred via the polymer cytoskeleton [18,90]. Among them, the use of stimuli-responsive biodegradable polymer nanoparticles to prepare drug delivery systems has great potential for controlled drug delivery [91,92]. It has been demonstrated that polymer degradation can be controlled by changing the external stimuli (e.g., pH, ultrasound, temperature, IR radiation, magnetic field, etc.), allowing stacked polymer nanoparticles to degrade in a controlled manner and release a drug on demand [90,93,94]. Qiu et al. successfully developed phosphatidylcholine–chitosan hybrid nanoparticles loaded with a gentamicin antibiotic and demonstrated that this synthetic system was able to inhibit the growth and membrane formation of Gram-positive and Gram-negative bacteria [95]. Studies have shown that by encapsulating vancomycin antibiotics in nanovesicles composed of long fatty acids grafted with hydrophilic polymers, these nanocarriers have the ability to self-assemble into spherical drug carriers and are effective against MRSA [96]. However, the polymer degradation products and clearance might cause potential toxicity as lipid-based nanoparticles typically have higher biocompatibilities than polymeric nanoparticles, which makes the application of polymeric nanoparticles in delivering antimicrobial agents a challenge. Hence, the field is still at an early development stage [19]. 

#### 2.2.2. Antimicrobial Cationic Polymeric Nanoparticles

Over the last decade, synthetic biodegradable antimicrobial cationic polymers have been a promising solution to combat bacteria. The cationic charges of these synthetic polymers selectively act and are attracted to negative-charged bacterial membranes on zwitterionic mammalian cell membranes, in a mechanism similar to natural antimicrobial peptides [97,98]. Antimicrobial cationic polymers have attracted tremendous attention owing to their facile synthesis in bulk quantities at much lower costs, broad spectrum efficacy of their antibacterial activity with membrane disruptive mechanism, as well as a low propensity for inducing bacterial resistance [99,100]. Of note are the natural antimicrobial peptide-mimicking antimicrobial cationic polymers brilacidin (ClinicalTrials.gov Identifier: NCT02324335) and LTX-109 (ClinicalTrials.gov Identifier: NCT01803035), which have completed phase 2 clinical trials.

The antibacterial mechanism of cationic polymers requires contact with a bacterial membrane’s outer surfaces, which induces a globally amphiphilic conformational change to sequester cationic and lipophilic side chains [101]. This property is known as facial amphiphilicity and is shown in Figure 3. The cationic subunits are responsible for interacting with the bacterial membrane, whereas the lipophilic side chains insert into bacterial membranes for subsequent membrane disruption. [102]. This leads to cytoplasmic leakage, membrane depolarization, lysis, and ultimately cell death, showing the promising antibacterial activity of these polymers [103]. It remains challenging to achieve proper facial amphiphilicity of cationic polymers. The majority of antimicrobial cationic polymers that are generated from uncontrolled polymeric self-assembly do not comprise truly facial amphiphilicity, which greatly affects antibacterial activity and can lead to nonspecific toxicity in mammalian cells [104]. Manipulation of the sequence of hydrophobic and hydrophilic subunits of antimicrobial polymers is an important factor in achieving facial amphiphilicity for antibacterial activity. A recent study combining vancomycin with the cationic polymer Eudragit E100 ^®^ (Eu) against *P. aeruginosa* showed that *P. aeruginosa* was eradicated within 3–6 h of exposure with this combination treatment [105]. Although bacterial envelope permeabilization and morphological changes after exposure to Eu were not sufficient to cause bacterial death, they allowed vancomycin to enter the target site, thereby enhancing the activity of an otherwise inactive vancomycin against *P. aeruginosa*.

The formulation of antimicrobial polymeric nanoparticles has overcome the aforementioned problems associated with antimicrobial polymers. The first antimicrobial polymer that self-assembled into cationic micellar nanoparticles by dissolution in water was reported by Nederberg [106]. A strong bactericidal activity of the cationic micellar nanoparticles was observed against MRSA and *Enterococcus faecalis* [106]. The polymeric nano-architecture was critical for effective bactericidal activity of the antimicrobial polymer molecules [106]. Unlike conventional antimicrobial polymers, the self-assembled antimicrobial polymeric nanoparticle does not require contact with the bacterial membrane for the formation of the secondary structure. It is hypothesized that the nanoparticle architecture increases the local concentration of cationic charge and polymer mass, leading to strong interactions between the polymer and cell membrane, which translate into effective antibacterial activities. Self-assembled antimicrobial polymeric nanoparticles have demonstrated minimal toxicity along with promising antibacterial activity, highlighting their potential in antibacterial applications and clinically relevant therapies [107,108,109,110]. Chin and colleagues reported a class of degradable guanidine-functionalized polycarbonates with a unique mechanism that does not induce drug resistance, which has great potential in the prevention and treatment of multidrug-resistant systemic infections [111]. The team optimized the structure of the polymer for treating multidrug-resistant *Klebsiella pneumoniae* pulmonary infections. In vivo experiments showed that the polymer backbone (pEt_20) self-assembles into micelles at high concentrations, which can alleviate lung infection with *K. pneumoniae* without causing damage to the major organs in mammals [112].

Another breakthrough study that benefits from this nanotechnology is star-shaped peptide polymer nanoparticles [113]. This is the first example of a synthetic antimicrobial polymer that efficiently kills colistin-resistant and multidrug-resistant Gram-negative pathogens, including *Acinetobacter baumannii*, *K. pneumoniae,* and *P. aeruginosa* [113]. The star-shaped peptide polymer nanoparticles eradicate these Gram-negative bacteria via destabilization and fragmentation of the bacterial outer membrane, disruption of cytoplasmic membrane, and induction of bacterial apoptosis [113]. Singh and her colleagues investigated the antimicrobial activities against clinical and drug-resistant strains (MDR-PA and MRSA) through indole-3-butyryl-polyethyleneimine nanostructured self-assembly in aqueous systems [114]. The amphiphilic indole-3-butyryl-polyethyleneimine polymer nanostructures have positively charged hydrophilic polyethyleneimine on the surface, while the hydrophobic indole-3-butyryl moiety is located inside the core, which showed enhanced antibacterial effects against all drug-resistant strains [114].

### 2.3. Carbon-Based Antimicrobial Nanomaterials

As a novel class of nanomaterials, carbon-based nanomaterials (CNMs) have received significant interest due to their remarkable properties including inherent antibacterial effects, extraordinary mechanical properties, excellent electrical conductivity and thermal conductivity, incredibly high surface area to volume ratios, photoluminescent and photocatalytic activities, and good stabilities [115]. These unique properties make carbon nanoarchitectures promising for a wide range of antibacterial applications including drug delivery, bone and tissue engineering, biosensors, photothermal therapy, and potential new antibacterial agents, which have been discussed elsewhere [116,117,118,119,120,121]. Due to its valency, carbon is able to form several allotropes that leads to a broad range of nanostructures of different dimensions, shapes, and properties, as depicted in Figure 4. 

Among carbon-based nanomaterials, the preferential use of graphene-based materials, especially graphene oxide (GO), for antibacterial applications is due to the following reasons: The highly oxygenated surface of GO, bearing hydroxyl, epoxide, diol, and carbonyl functional groups, provides a versatile platform for drug delivery applications or further functionalization [122]. The good aqueous solubility of GO makes it suitable for in vivo antibacterial applications compared with poorly water-soluble fullerenes and nanotubes [20,123]. Another advantage of graphene oxide is its ability to act as a barrier or overlay, delaying and controlling the release of biomolecules over time [124,125,126]. Lastly, GO synthesis can be devoid of any metallic impurities and these materials can exhibit tolerable toxicity [127,128].

#### 2.3.1. Graphene Oxide as an Antimicrobial Delivery Nanocarrier 

The use of graphene oxide as a nanocarrier for drug delivery has received significant attention for several reasons, such as its good biocompatibility with tolerable toxicities [129,130]. In addition, the ease of functionalization provides possibilities in synthesizing novel functional nanohybrids or nanocomposites for specific purposes including targeted drug delivery, as shown in Figure 5 [131]. The extremely large surface area coupled with a two-dimensional planar structure provides a huge drug-loading capacity. In fact, a significant rate of drug loading has been reported previously with a GO-based delivery system [21]. The high mechanical and chemical stability of GO makes it particularly suitable for different delivery environments [132]. Currently, GO nanoparticles have been experimentally used in various biomedical applications including gene delivery [133], drug delivery [134,135], photodynamic therapy [136], anticancer therapies [137,138], and antibacterial therapies [139]. Nevertheless, studies of GO-based delivery systems with bacterial infection are still at a preliminary stage, involving investigations in antibiotic absorption efficacy and the respective in vitro antibacterial activity [140,141]. In a recent study, polyethylene-glycol-functionalized GO nanoparticles loaded with *Nigella sativa* seed extract were tested as a drug delivery system to disrupt bacteria by penetrating bacterial nucleic acid and cytoplasmic membranes, successfully demonstrating potential antibacterial activity against *S. aureus* and *Escherichia coli* [142]. Pan et al. used GO as a carrier to load N-halamine compounds, which not only displayed an antibacterial effect against *S. aureus* and *E. coli*, but also had slow-release properties and good storage stability [143]. In general, with the aforementioned advantages conferred by GO-based delivery systems, the potential of GO as a delivery platform for antimicrobial agents should not be neglected.

#### 2.3.2. Graphene Oxide with Inherent Antibacterial Properties

GO has received tremendous attention as a novel antibacterial agent compared with its role as a delivery nanocarrier in antibacterial applications due to its broad-spectrum antibacterial activity and low cytotoxicity at low concentrations [144]. GO has been reported to exhibit strong antibacterial activity against a variety of Gram-positive and Gram-negative bacteria, such as *E. coli*, *S. aureus*, *E. faecalis, P. aeruginosa*, and *Candida albicans* [22,139,145,146,147,148,149]. In addition, Di Giulio et al. also reported significant antibiofilm efficacy against biofilms produced by *S. aureus, P. aeruginosa*, and *C. albicans* [150]. Recently, attention has also been given to GO-based combination antibacterial therapies. The ternary nanocomposites obtained by combining GO with hydroxyapatite and copper oxide have inhibitory effects on Gram-negative *E. coli* and Gram-positive *S. aureus* [151]. Innovative bionanomaterials composed of GO, agarose, and hydroxyapatite have also shown the ability to significantly reduce *S. aureus* [152].

The strong antibacterial activity of GO is associated with both physical and chemical damages. The physical interactions of GO with bacteria that are reported to date include interactions via direct contact of its sharp edges, lipid extraction, bacteria isolation from their nutrient environment by wrapping, and photothermal/photocatalytic effects owing to the semiconductor properties of graphene [139,153,154,155,156]. Interestingly, the mechanism of action of GO on Gram-positive and Gram-negative bacteria appears to be distinct. Pulingam et al. reported that cell entrapment via mechanical wrapping was mainly observed for the Gram-positive bacteria *S. aureus* and *E. faecalis*, whereas with Gram-negative bacteria *E. coli* and *P. aeruginosa* it was observed that membrane rupture due to physical contact [22] was the predominant mechanism. Chemical damages are additionally caused via oxidative stress and generation of reactive oxygen species (ROS) and charge transfer, thereby inhibiting bacterial metabolism, disrupting cellular functions, causing inactivation of intracellular and subcellular proteins, and inducing lipid peroxidation, leading to cellular inactivation [147,157,158,159]. Zhang et al. recently highlighted electrical conductivity as a key property of GO that may be underestimated in terms of its antibacterial activity role [160]. Research by Chong et al. proposed that sunlight irradiation could increase the antibacterial activity of GO due to enhancing the electron transportation of antioxidants [161]. GO’s diverse physicochemical properties including sheet size, shape, number of layers, surface charge, defect density, and the presence of surface functional groups and oxygen content have a strong impact on its antibacterial activity and biological performance [116,162]. However, the physicochemical properties related to antibacterial activity are not fully elucidated yet. Deepening the understanding of the physicochemical properties related to antibacterial activity is a crucial step in designing GO-based nanomaterials for optimized antibacterial activity. Together, these studies provide important insights into the way forwards.

## 3. Antibacterial Inorganic Nanomaterials

Inorganic nanomaterials do not contain either carbon or hydrogen atoms that are associated with biological matter. As an alternative, inorganic nanomaterials comprise metallic and non-metallic elemental compounds that have weak intermolecular interactions which form nanostructures with higher dimensionality. Among two classes of the inorganic nanomaterials, metallic inorganic nanomaterials (Ag, Au, Zn, Cu, Bi) have attracted significant attention over non-metallic nanomaterials (S, Si, B, Te and Se). This is particularly due to the inherent water insolubility of the non-metallic inorganic nanomaterials that restricted their use in antibacterial applications. Therefore, this review primarily focuses on the development of inorganic metallic nanomaterials in antibacterial applications. Other inorganic nanomaterials such as fluoride in oral treatment has also been researched for a long time. For instance, in the prevention of dental caries, the addition of fluoride has a significant antibacterial effect on *Streptococcus mutans*, *Lactobacillus acidophilus*, *E. faecalis*, *Actinomyces naeslundii,* and *Parvimonas micra* [163,164,165,166].

Inorganic metallic nanomaterials do not readily self-assemble into 1D nanowires, nanotubes, nanoribbons, and 2D nanowalls and nanofilms [167]. Therefore, complicated synthesis methodologies are required to promote the “growth” of inorganic metallic nanomaterials into nanostructures with higher dimensionality [168,169]. Zero-dimensional metal and metal oxide nanoparticles are the most popular candidates that can readily be synthesized for antibacterial applications, which are discussed in Section 3.1 and Section 3.2 below. 

### 3.1. Metal Nanoparticles

Metal nanoparticles are the most promising candidate in this class of materials with inherently strong antibacterial activities amongst the nanomaterials. A summary of the possible bactericidal effects of metal nanoparticles on different bacteria is shown in Table 3. Researchers reviewed a variety of metal nanoparticles including silver, gold, copper, zinc, and super-paramagnetic iron which demonstrated promising antibacterial effects, with silver nanoparticles (AgNPs) being the most effective against bacteria [169,170,171,172]. AgNPs exhibit bactericidal activity at concentrations well below their cytotoxicity and exhibit synergistic antibacterial efficacy with conventional antibiotics when used against MDR bacteria [173,174,175,176].

The antibacterial mechanisms of AgNPs are still poorly understood despite extensive studies [171,176]. Currently accepted antibacterial mechanisms include cell wall penetration and membrane damage, toxicity associated with metal ion release, and induction of oxidative stress [177,178,179,180]. AgNPs have already been used in various biomedical and antibacterial applications and products, including surface coatings on medical devices, topical treatments, wound dressings, dental fillings, personal care products with sanitizing effects, disinfectants, and detergents [180,181]. A recent study demonstrated the potential of AgNP-containing disinfectants as active ingredients for disinfecting surgical masks, effectively improving mask protection by inhibiting the growth of *E. coli*, *K. pneumoniae*, and *S. aureus* [23]. Over the past few decades, the AgNP market has been growing steadily, with an estimated annual production of more than 500 tons of nanoparticles, which also reflects the widespread interest of AgNPs [182]. 

Despite the promising antibacterial effects and the wide use of metal nanoparticles in different applications, metal nanoparticles suffer from several drawbacks. The potential toxicity of metal nanoparticles affects the basic functioning of mammalian cells as metal nanoparticles or released metal ions via direct uptake from mammalian cells [24]. Colloidal metal nanoparticles tend to aggregate over time [25]. The aggregation along with increased particle size reduces their peculiar properties at the nanoscale, including their antibacterial activities. Bacteria have the ability to develop resistance to metal nanoparticles by using adhesive flagellin [183]. Phenotypic changes of adhesive flagellin production triggers the aggregation of metal nanoparticles, thereby reducing their antibacterial activity. Metal nanoparticles are potential environmental hazards and difficult to recover or deactivate in solid-waste incineration plants or wastewater treatment systems [184,185,186]. To overcome the above limitations, metal nanoparticles could be incorporated into other nanomaterials to form nanocomposites, which show a greater dispersion of metal nanoparticles, improved antibacterial activity, and reduced toxicity [187,188,189]. This will be further discussed in Section 4 below.

### 3.2. Metal Oxide Nanoparticles

Metal oxide nanoparticles offer another alternative promising solution against MDR bacteria. A variety of metal oxide nanoparticles including titanium dioxide, zinc oxide, magnesium oxide, copper oxide, and aluminium oxide have been demonstrated to exhibit antibacterial effects [215], which will not be discussed in detail here. Zinc oxide (ZnO) nanoparticles are the most-established candidate of these metal oxide nanoparticles due to the following reasons: ZnO is one of the most important metal oxide nanoparticles with widespread applications [26] and worldwide production is up to 1 million tons per year [216]. ZnO can be easily biodegraded and absorbed in the body and has been listed as a Generally Recognized as Safe (GRAS) material by the FDA. ZnO nanoparticles have a higher biocompatibility and lower toxicity than other metal oxide nanoparticles [217,218]. The semiconductor properties of ZnO nanoparticles with a wide band-gap energy readily absorb ultraviolet (UV) light. This allows them to act as potential photosensitizing agents for various antibacterial applications [27]. ZnO nanoparticles exhibit multiple antibacterial mechanisms, as described by Figure 6. Current postulated mechanisms include photo-triggered production of ROS [219] and Zn^2+^ ions mediating a poisoning effect [220]. In a study by Azam et al., the potential of ZnO as an antibacterial agent was demonstrated against Gram-positive bacteria (*B. subtilis*, *S. aureus*) and Gram-negative bacteria (*P. aeruginosa*, *Campylobacter jejuni*, *E. coli*) [213]. In another study, ZnO nanoparticles were also shown to be significantly inhibitory against Gram-positive (*S. aureus*) and Gram-negative (*E. coli* and *P. aeruginosa*) strains [221].

The aggregation of metal oxide nanoparticles over time remains a problem, which limits their use in vivo applications [223]. One approach to this problem is to disperse metal oxide nanoparticles into a polymer matrix, forming polymer–metal oxide nanocomposites. Polymer–metal oxide nanocomposites have been widely investigated and applied in the textile and polymer industries for various antibacterial applications, which will be discussed in the following section. [224]. 

## 4. Nanocomposite/Nanohybrid Antibacterial Materials

Nanocomposites or nanohybrids are a novel class of multiphase materials that exhibit a hierarchical structure, where one phase of the material has at least one dimension in the nanometer range [225]. They have attracted significant attention due to their unprecedented properties compared with their mono-constituent parts, largely attributed to strong reinforcing effects of additional materials. Currently, synthesizing a variety of nanocomposites with unprecedented physical properties and enhanced antibacterial activities is the main focus in the field over the last 10 years. Despite the large volume of studies on nanocomposites, the understanding of the structure–property–activity changes remains in its infancy [226]. It is important to understand the mechanisms behind the property’s changes within nanocomposites in order to design materials with enhanced improvements in the desired properties for a specific purpose. Section 4.1, Section 4.2, Section 4.3, Section 4.4 and Section 4.5 discusses the development and potential applications of the nanocomposite antibacterial materials. Of note, graphene oxide is especially emphasized in the section below due to its versatility to form nanocomposites with organic and inorganic metallic nanomaterials, respectively. 

### 4.1. Polymer–Metal Nanocomposite Nanoparticles

Polymer–metal composite nanoparticles are another promising solution to achieve a greater dispersion of metal nanoparticles and prevent metal nanoparticle aggregation. Polymer–metal composite nanoparticles comprise a metal nanoparticle core surrounded by a polymer shell with the alkyl tail arranged toward the surrounding environment. Polymer–metal composites are essentially insoluble in water and the colloidal stability of the polymer–metal composite nanoparticles in aqueous environments has a huge potential for in vivo antibacterial therapies [227]. The polymer macromolecular matrix acts as a reaction chamber for metal nanoparticle synthesis, a capping agent to prevent nanoparticle aggregation and as a scaffold for nanoparticle immobilization [228]. Moreover, synergies between the polymer and the metal nanoparticles confer the nanocomposite with unprecedented performance and improved antibacterial properties [229]. Tamayo et al. summarized the synthesis, properties, and recent applications of polymer composites with metal nanoparticles [230]. The incorporated metal nanoparticles are focused on the use of gold (AuNPs) and silver (AgNPs) due to their antimicrobial properties, catalytic activity, and conductivity properties enabling a wide range of applications. 

#### 4.1.1. Development of Synthesis Approaches for Polymer–Metal Nanocomposites

Both in situ and ex situ approaches can be employed to synthesize polymer–metal composite nanoparticles and polymer-matrix metal nanocomposites. In the last decade, several studies have been conducted to develop and improve synthetic methods at higher efficiencies resulting in improved antimicrobial outcomes.

By using in situ methods, the precursor of the nanoparticle is required to be dispersed in a monomeric solution before polymerization. The metal ions can be reduced into the polymer matrix or simultaneous metal ion reduction and polymerization can occur [230]. In general, these in situ reduction methods need a relatively long time to produce nanocomposite films. Kazuhiko et al. developed a rapid and scalable synthetic method exploiting use of a mid-infrared laser, CO_2_ laser, at 10.6 µm without the use of reducing agents [231]. The polymer film (polyvinyl alcohol (PVA) or polyethylene glycol (PEG)) containing Ag ions were coated on a glass substrate and then the CO_2_ laser was used to heat the substrate. Subsequently, the thermal energy was absorbed by the polymer film, causing Ag ion reduction. Eventually, Ag-PVA or Ag-PEG nanocomposite films were formed in several seconds. This process is industrially scalable by increasing the power of the CO_2_ laser. 

Ultrasound is a promising tool to be applied in for the in situ production of polymer–metal nanocomposites. Ultrasound radiation was employed as a homogenizing tool to fabricate composites with homogeneously dispersed metal nanoparticles [232]. The ultrasound was used to disperse organic liquids of polymerizing monomer (pyrrole) in the aqueous solution of the oxidizer (Ag^+^ or AuCl^−^). The aqueous solution was placed in an ultrasonic chamber and droplets of the organic solution were added continuously until achieving a volume ratio of 4:1. After polymerization, the nanocomposites could be obtained at the liquid–liquid interface [232]. Wan et al. used ultrasound as both an initiating and reducing agent in the nanocomposite preparation process, shown in Figure 7 [233]. Tertiary amine-containing polymeric nanoparticles were produced by ultrasound-initiated polymerization-induced self-assembly (sono-PISA), following which, the metal ions (Au and Pd) were reduced in situ by radicals generated via the sonolysis of water, forming polymer–metal composites.

In contrast, metal nanoparticles are synthesized before they are incorporated into the polymer via an ex situ method. The subsequent deposition of the nanoparticles into the polymer can exploit processes such as melt compounding or solution blending [230]. However, a significant drawback exists when using ex situ methods, which is the fact that the nanoparticles are not optimally distributed in the polymer.

#### 4.1.2. Synergistic or Combined Antibacterial Effects When Using More Than Just a Metal Nanoparticle Agent

A combination of metal nanoparticles and cationic polymers also facilitates the enhanced antibacterial activity of a composite nanoparticle, possibly due to the synergism between the antibacterial mechanisms from two different nanomaterials [234,235,236,237]. Nanoparticle formation was also expected to increase the local density of cationic polymer, leading to stronger binding on the negatively charged bacterial membranes [234]. This polyvalent interaction between cationic polymers and bacterial membranes is followed by the synergistic antibacterial mechanisms, including bacterial membrane disruption, internalization of composite nanoparticles, inhibition of intracellular enzymatic activity, and eventual cell death [234]. Imidazole-capped chitosan–gold nanocomposites exhibited enhanced antimicrobial activity to eradicate staphylococcal biofilms in a rabbit wound infection model [238]. The antibacterial mechanism of the composite nanoparticles also involved the binding of cationic polymer to the bacterial surface, and the subsequent synergistic effects from the gold nanoparticles, imidazole, and the chitosan polymer to strongly eradicate the biofilm [238]. 

Polymer–metal composite nanoparticles could potentially solve the environmental hazards associated with many metal nanoparticles. Richter et al. postulated that a metallic core is not necessary for the antimicrobial action [187]. Instead of synthesizing the entire nanoparticle of metal, the composite nanoparticle can be produced by infusion with a minimum amount of silver ions to the biodegradable lignin core, followed by surface functionalization with a layer of cationic polyelectrolyte [187]. The resulting composite nanoparticles exhibited broad-spectrum bactericidal activity against *E. coli*, *P. aeruginosa,* and a quaternary-amine-resistant *Ralstonia*. This is attributed to the enhanced binding to bacterial membranes by the polyelectrolyte shell and the synergistic antibacterial activity between the silver ions and the polyelectrolyte. The required silver ions were ten times lesser than when using conventional silver nanoparticles. The gradual diffusion of silver ions from the silver-infused lignin core composite nanoparticles into water will rapidly lose their post-utilization activity and be biodegradable in the environment after disposal [187].

#### 4.1.3. On the Potential Clinical Use of Antibacterial Polymer-Matrix Metal Nanocomposites 

Biocompatible and safe antibacterial materials are constantly sought to avoid inflammatory syndromes in patients. The formulation of polymer–metal composite nanoparticles generally improves their biocompatibility and reduces the toxicity associated with metal nanoparticles owing to protection by the polymeric shell. For instance, the viability of NIH3T3 cells was not affected at a dosage exceeding 20 times that of the minimum inhibitory concentration (MIC) of the polymer–silver composite nanoparticle, showing the low toxicity of these materials to mammalian cells [234]. Lu et al. further demonstrated that imidazole-capped chitosan–gold nanocomposites did not display hemolytic activity and significant toxicity towards L929 cell line [238]. Pryjmaková et al. modified the surface of polyethylene naphthalate (PEN) by a 248 nm KrF excimer laser and subsequently, Ag and Au nanowires were incorporated onto the modified PEN surface by vacuum evaporation [239]. The resulted nanocomposites displayed antibacterial effects against Gram-negative bacteria (*E. coli*) and Gram-positive bacteria (*S. epidermidis*) via a 24-hr incubation drop plate test and were suggested as a non-toxic material by a WST-1 cytotoxicity test. In addition to the complete eradication of the biofilm, accelerated wound healing by a composite nanoparticle was demonstrated in a rabbit model [238]. These studies demonstrate the great potential of composite nanoparticles as novel antibacterial agents against bacterial infections.

### 4.2. Polymer-Matrix Metal Oxide Nanocomposites

Fine dispersions of metal oxide nanoparticles can be achieved by forming polymer-matrix metal oxide nanocomposites in a manufacturing process similar to polymer-matrix metal nanocomposites. There is an increasing interest in using metal oxide nanoparticles to replace metal nanoparticles for synthesizing polymer-matrix metal oxide nanocomposites [240,241,242,243]. Metal oxide nanoparticles, especially zinc oxide nanoparticles, are more desirable than metal nanoparticles as nanofillers in forming polymer-matrix nanocomposites [244,245,246,247]. Zinc oxide nanoparticles feature new UV-absorption and photosensitizer characteristics, with higher biocompatibilities and lower toxicities than metal nanoparticles [218,248]. Furthermore, the low cost of production and high stability of zinc oxide nanoparticles present advantages over conventional metal nanoparticles, even in extreme synthesis conditions [249]. The advantages of metal oxide nanoparticles as nanofillers in forming polymer-matrix metal oxide nanocomposites has open a new avenue for research into novel bio-nanocomposites for use as antimicrobial surfaces in various antibacterial applications, which will be depicted in following sections.

Polymer-matrix metal nanocomposites exhibit several advantages. Polymer flexibility allows the final product to be fabricated into complex structures or forms for various antibacterial applications, of which the details are shown in Table 4. The polymeric matrix immobilizes metal nanoparticles and prevents their aggregation, thus extending the antibacterial activity of the metal nanoparticles [228]. Localized release of metal nanoparticles to the desired application site can also be achieved, reducing in vivo toxicity and the environmental hazards caused by undesirable release of metal nanoparticles [250]. Synergistic antibacterial activity between metal nanoparticles and polymers is obtained with inherent antibacterial activities [229]. The strong interfacial binding and intermolecular interactions between the well dispersed metal nanoparticles and the polymer matrix further enhance the mechanical properties of the polymer, including the tensile strength, Young’s modulus, yield stress, and ductility [251].

#### 4.2.1. Development of Synthesis Approaches for the Industrial Production of Polymer-Matrix Metal Nanocomposites

A prerequisite to the aforementioned advantages exhibited by polymer-matrix metal nanocomposites is the formation of homogenous dispersions of metal nanoparticles in the polymer matrix without metal nanoparticle aggregation [262]. The delicate synthesis conditions to meet this prerequisite is often time-consuming, laborious, and difficult to be industrialized [263]. Therefore, recent studies have focused on developing facile and convenient synthesis approaches, aiming to produce polymer-matrix metal nanocomposites on industrial scales [228,250,264]. Other recent review papers have summarized developments, so we will not discuss these in detail [240,241,242]. 

For instance, Tran et al., developed a simple one-pot synthesis method in producing polymer matrix silver nanocomposites [228]. The ionic liquid medium, butylmethylimmidazolium chloride, was utilized as the only reaction medium for dissolving the biopolymer keratin and cellulose, and reduction of a silver ion precursor in the polymeric matrix. The synthesized polymer-matrix silver nanocomposite was found to retain the enhanced mechanical strength by cellulose and controlled release of silver nanoparticles by keratin, with a homogenous dispersion of silver nanoparticles. At 0.48 mmol of silver content, the nanocomposite demonstrated good biocompatibility and excellent antibacterial activity against *E. coli*, *P. aeruginosa*, MRSA, and vancomycin-resistant *E. faecalis* (VRE). An in vitro release assay demonstrated that less than 0.02% of the silver nanoparticles were released from the nanocomposite even after 7 days of soaking in solution, indicating good immobilization of silver nanoparticles using this simple one-pot synthesis method [228]. 

A scalable approach was recently developed to produce a silver-nanoparticles-doped nanoclay–polylactic acid composite nanocomposite, which involved doped nanoclay with minimal alteration to the fabrication processes and industry standard equipment [264]. Loading the nanoclay can significantly reduce the affinity of the nanocomposites for bacterial adhesion. With this synthesis method, only a 0.1 wt % of silver loading content was required to have satisfactory antibacterial activity. *S. aureus* and *E. coli* numbers were reduced by 91.3% and 90.7% after 48 h of incubation. The material costs associated with the silver-loading content is dramatically reduced compared with other studies which utilize at least 1 wt % of silver nanoparticles in the polymer-matrix metal nanocomposite to achieve a 90% reduction in bacterial numbers [265,266]. This is a great advantage for industrial production, in which the high costs associated with higher loading amounts of metal nanoparticles is a considerable problem. In addition, 3D printing is a promising method to produce metal oxide nanocomposites, with the advantages of keeping the integrity and functionality of the materials and reduce waste from traditional manufacturing methods [267].

#### 4.2.2. The Application of Polymer-Matrix Metal Oxide Nanocomposites as Self-Sterilizing Antimicrobial Surfaces in Healthcare Environments 

There are some recent literature reviews on the application of polymer-matrix metal oxide nanocomposites as self-sterilizing antimicrobial surfaces in healthcare environments which will not be repeated here [268,269,270]. The antibacterial and photosensitizing activity of ZnO nanoparticles has been well exploited in the absence or presence of light irradiation [271]. With the exploitation of their light absorption characteristic, ROS are produced from ZnO nanoparticles to act on bacteria, leading to a self-cleaning or self-sterilizing effect of the polymer-matrix zinc oxide nanocomposites [272]. For instance, Sehmi et al. and Ozkan et al. have developed self-sterilizing surfaces that was coupled with light-activated photodynamic therapy in killing bacteria [272,273]. Both studies showed the polymer matrix zinc oxide nanocomposites demonstrated lethal photosensitization of *E. coli* and *S. aureus* under white light irradiation that has a similar light intensity to that in a clinical setting [272,273]. This could potentially lower the rates of healthcare-associated infections by eliminating bacterial transfer in healthcare environments.

#### 4.2.3. Wound Healing Applications of Polymer-Matrix Metal Oxide Nanocomposites 

Polymer-matrix metal oxide nanocomposites are an attracting candidate in wound healing applications. Gobi et al. summarized the recent applications of nanocomposites in wound dressings [274]. A novel polymer-matrix metal oxide nanocomposite comprising a castor oil polymeric matrix reinforced with a chitosan-modified ZnO nanocomposite was recently developed [275]. This novel bio-nanocomposite showed enhanced mechanical properties, porosity, water absorption, hydrophilicity, water vapor transmission rate, and oxygen permeability [275]. These enhanced properties are important for wound healing, which provides porosity to absorb wound exudates and water, enables a moist wound healing environment, a cooling effect for pain alleviation, and gases to exchange for ventilation. ZnO nanofillers also enhanced the antibacterial activity and keratinocyte migration of a polymer-matrix zinc oxide nanocomposite, leading to stronger antibacterial activity that prevented the reoccurrence of a bacterial infection and promoted healing [276]. Recent research has reported a prepared nanocomposite consisting of a Lawsone-loaded o-carboxymethyl chitosan and ZnO which was evaluated against bacterial strains such as *Salmonella*, *S. aureus*, *P. aeruginosa*, and *E. coli* [277]. This prepared nanocomposite tended to prevent the evolution of these harmful bacteria compared to an o-carboxymethyl chitosan or nano-zinc oxide alone, further supporting this advantageous strategy of using a polymer-matrix ZnO nanocomposite in wound dressings. Finally, a polymer-matrix ZnO nanocomposite demonstrated promising in vivo efficacy, biodegradability, cytocompatibility, and promoted cell attachment on the material [278,279]. Taken as a whole, polymer-matrix metal oxide nanocomposites could possibly satisfy all the required standards as wound materials, highlighting their huge potential in wound healing applications. 

#### 4.2.4. Food Packaging Applications of Polymer-Matrix Metal Oxide Nanocomposites

For food packaging applications, the addition of ZnO nanoparticles as nanofillers to biodegradable polymeric materials greatly enhances the physiochemical properties and antibacterial activities of the resulting bio-nanocomposites to protect the environment [270,280,281]. ZnO nanoparticles create a barrier effect to hinder the diffusion of the decomposition products from the polymer matrix to the gas phase, which further improve polymer thermal stability and avoid thermal degradation under the wide polymer melt processing window [282]. In addition, ZnO nanoparticles act as a nucleating agent in raising the crystallinity level of the polymer matrix [283]. The combined effect of the increased crystallinity of the polymer along with the barrier effect of ZnO nanoparticles creates a highly tortuous path for the gases, water vapors, and organic compounds [284]. 

The barrier properties to gases, water vapors, and organic compounds subsequently improve the product quality and shelf life by blocking the diffusion of moisture and oxygen. Mechanical performance such as stiffness, glass transition temperature, tensile strength, and toughness is also enhanced because of the strong polymer matrix–ZnO nanofiller interactions [282,285]. The antibacterial and UV-absorption properties of ZnO nanoparticles inhibit the growth of food-borne pathogens and prevent the photo-oxidative degradation of food, respectively [286]. Taken together, polymer matrix–zinc oxide nanocomposites are promising materials to be used as cutlery, overwrap films, and containers in preventing growth of food-borne pathogens and achieving good quality packaged food with extended shelf-life.

### 4.3. Graphene Oxide–Metal Nanocomposites

As mentioned in Section 3, the shortcomings of metal nanoparticles limit their potential for medical applications. To overcome these issues, many nanocomposites composed of metal nanoparticles and graphene have been prepared experimentally and studied against various bacterial strains [287,288,289,290,291,292]. Some of the available literature reviewed the development of graphene–metal matrix nanocomposites which will not be repeated here [293,294,295,296]. Among them, a combination of graphene oxide and silver nanoparticles to form nanocomposites has attracted a lot of attention as antibacterial agents in antibacterial therapies, since Ag and its compounds have been used since the time of the ancient Egyptians. The antibacterial and antiviral properties of Ag, Ag ions, and Ag-based compounds have been thoroughly researched [202,297,298]. With the incorporation of GO as the supporting matrix, silver nanoparticles could be dispersed in aqueous solution while minimizing the aggregation problem that would otherwise greatly affect the antibacterial activity of the silver nanoparticles [299]. The large surface area and abundant functional groups on the basal plane of GO allows GO to interact with silver ions or silver nanoparticles through electrostatic interactions, charge–transfer interactions and physical absorption [300]. This allows GO–Ag nanocomposites to be synthesized through loading of pre-synthesized silver nanoparticles into GO (ex situ approach) or via reduction of silver ions in a graphene matrix to form silver nanoparticles in situ [301,302,303,304,305]. 

#### 4.3.1. Development of Synthetic Approaches for Improving the In Vivo Performance of Graphene Oxide–Metal Nanocomposites 

The synthesis process of GO/reduced GO (rGO)–Ag nanocomposites involves harsh conditions, as well as highly toxic reducing agents and organic solvents, which minimizes their use in biomedical applications, which is an unmet area of need that requires more exploration [306,307]. Despite the good dispersity of GO–Ag in aqueous solution, it has been discovered that GO aggregates irreversibly in physiological solutions over time [308]. This greatly affect its bioavailability, significantly weakening its antibacterial efficiency and long-term effectiveness. In an effort to solve the problem, the (polyethylene glycol) PEGylation of GO was carried out for long term antibacterial activity and stability of GO–Ag in physiological solution [309]. The PEGylated GO–Ag nanocomposite remained stable in a series of complex media over one month and resisted centrifugation (Figure 8). In contrast, non-PEGylated GO–Ag aggregated to varying degrees in the media after 1 h, and complete precipitation was observed after 1 week of equilibration. GO−PEG−Ag nanocomposites displayed remarkable long-term antibacterial activity after 1 week of storage in physiological saline, preserving >99% antibacterial activity against *S. aureus* and >95% antibacterial activity against *E. coli*. GO−PEG−Ag inhibited bacterial growth in nutrient rich Luria–Bertani (LB) broth for at least one week, and the repeated usage of GO−PEG−Ag up to three times did not reduce the antibacterial efficacy. In contrast, unmodified GO−Ag exhibited a >60% decline in antibacterial activity after 1 week of storage in physiological saline. This study provides a direct solution for the synthesis of homogenously dispersed and stable GO–Ag nanocomposites under physiological conditions. This result was also confirmed by a subsequent study, in which ternary hybrids of PEG-functionalized GO with Ag nanoparticles exhibited excellent bactericidal effects against *E. coli*, and it was found that those with smaller Ag nanoparticles (8 nm) showed better antibacterial activity than those with larger nanoparticles (50 nm) [310]. Furthermore, modification of GO with polyethyleneimine polymers dramatically enhanced the long-term antibacterial activity and stability of the GO–Ag nanocomposite [311]. In addition to this, Parandhaman and his colleagues recently designed GO–Ag nanocomposites functionalized with the natural antimicrobial peptide poly-L-lysine with remarkably improved stability and adhesion to *S. aureus* biofilms [312]. Notably, poly-L-lysine functionalization prevented the leaching of anions, thereby reducing the cytotoxicity of the graphene–silver nanocomposites. In order to obtain nanomaterials with long-term and stable antibacterial activity, a facile and green method has also been proposed to prepare AgNPs/polymer/GO composites with catalytic and antibacterial activities via the incorporation of furan-functional poly(styrene-*alt*-maleic anhydride) [313].

#### 4.3.2. Potential of Graphene Oxide–Metal Nanocomposites for In Vivo Therapies

In terms of biological activity, GO–Ag nanocomposites have been shown to demonstrate synergistic antibacterial activities against planktonic bacteria and biofilms, with low cytotoxicity and good biocompatibilities [314,315,316,317]. The enhanced antibacterial activity of GO–Ag nanocomposites is often ascribed to the synergistic activity of GO and Ag; however, the full antibacterial mechanism remains to be elucidated [299]. Malik et al. have also demonstrated that GO–Ag nanocomposites exhibit significantly enhanced growth inhibition of *E. coli*, *S. aureus*, and *P. aeruginosa* relative to silver nanoparticles alone [291]. Recent studies have accomplished the surface functionalization of GO and Ag nanoparticles by using lantana plant extract, and the results also affirmed the potential of GO–Ag nanocomposites as antibacterial agents against biological pollutants [290]. The negatively charged oxygen-containing groups of graphene oxide can absorb Ag ions through electron absorption, which can improve the confinement of Ag nanoparticle agglomeration and burst release, and synergistically enhance their antibacterial properties [292]. Intriguingly, GO–Ag nanocomposites exhibit species-specific bactericidal mechanisms, with cell wall disruption being observed against *E. coli* and inhibition of cell division against *S. aureus* [318]. 

The promising physical properties of GO, along with its synergistic activity with Ag nanoparticles, hold great potential for a targeted nanocomposite system [319]. A photothermal nanocomposite was produced which was composed of hyaluronic-acid-coated Ag nanoparticles that were integrated with GO [319]. Hyaluronic-acid-coated Ag nanoparticles confer additional protection by preventing the release of metal ions to surrounding mammalian cells. Upon encountering bacteria that secrete hyaluronidase, such as *S. aureus*, hyaluronic acid is degraded, followed by interaction of the GO–Ag nanocomposite and bacteria to further enhance the antibacterial action. Together with the photocatalytic characteristic of GO, local photothermal therapy under light irradiation could be achieved to further enhance the antibacterial activity of the GO–Ag nanocomposite [291]. 

#### 4.3.3. Potential of Graphene Oxide–Metal Nanocomposites to Reduce Membrane Biofouling Issues for Water Decontamination and Filtration 

The intrinsic characteristics of GO, including its availability as single-atomic-thick sheets, high hydrophilicity, extraordinary electrical, thermal, mechanical, structural properties, and low systemic toxicity could potentially reduce membrane biofouling issues for water decontamination or filtration [320,321]. GO–Ag nanocomposites are also a promising membrane surface modifier that contributed to enhanced membrane hydrophilicity, wettability and permeability, and good water influx [322,323,324,325]. Surface modification of membranes using GO/Ag nanocomposites exhibited stronger antimicrobial activities than AgNP-modified membranes and GO-modified membranes, without significantly altering the membrane transport properties [326]. The feasibility of using GO–Ag nanocomposites in membrane regeneration for a long-term anti-biofouling effect was demonstrated by conducting, in situ, the Ag-formation procedure to regenerate AgNPs on GO–Ag-modified membranes [327]. This potentially solves the problem of weakening biofouling properties of the functionalized GO–Ag nanocomposite membranes over time due to constant leaching of silver ions throughout the process. More complex membranes containing GO, Ag, and metal-organic frameworks (MOFs) in PES were prepared for water treatments, exploiting the synergistic effects of graphene oxide and silver to enhance the anti-biofouling properties of the membranes [328]. In conclusion, the metal oxide/graphene nanocomposites exhibit enhanced antibacterial properties under visible light irradiation and have great potential as photocatalysts in the field of water purification [321]. Nevertheless, more studies are required to examine the long-term usage, membrane reusability, and regeneration potential of functionalized GO–Ag nanocomposite membranes.

### 4.4. Graphene Oxide–Polymer Nanocomposites

Graphene oxide–polymer nanocomposites exhibit enhanced antibacterial activity, biocompatibility, hemocompatibility, hydrophilicity, and stability, compared with the polymeric based nanomaterials [329,330,331]. In addition, GO can greatly reinforce the mechanical properties of GO–polymer nanocomposites, including their breaking strength, Young’s modulus, compressive strength, flexural strength, and tensile strength [332,333,334]. The reinforcing effect is usually explained via strong interactions and bonding between the homogenously dispersed GO and polymeric components [332,333]. 

The alignment of GO sheets on the polymer film has been suggested to greatly affect the antibacterial activity of the resulting graphene oxide-based polymer nanocomposite [335]. Lu et al. synthesized graphene oxide–polymer nanocomposites by aligning GO in planar, vertical, and random orientations with the aid of a magnetic field (Figure 9A). GO was then immobilized by cross-linking with the surrounding polymer matrix, followed by oxidative etching to expose GO on the surface (Figure 9B). The vertically aligned GO nanosheets on the polymer film exhibited enhanced antibacterial activity compared with the random and horizontal orientations. Mechanistic examinations revealed that direct, edge-mediated contact with bacteria was the major mechanism in causing a greater physical disruption of the bacteria membranes (Figure 9C) [335]. Subsequently, greater levels of intracellular electron donors, for instance, glutathione, would release into the external environment upon membrane disruption, favoring GO to induce antibacterial activity via an oxidative stress mechanism [335]. This study highlights the importance of GO alignment and provides direct implications for the designing of GO–polymer nanocomposite films with enhanced antibacterial activities.

The abundant functional groups present on GO–polymer nanocomposites provide various interactions with nanoparticles, creating GO–polymer-based metal nanocomposites with superior characteristics. For instance, GO–chitosan nanocomposites have been demonstrated to act as both nucleation sites for calcium phosphates mineralization and absorption sites for nanoparticles [336]. The resulting GO/chitosan nanocomposites comprise micro- and nanohierarchical porous structures that allow cell attachment and proliferation after biomineralization [336]. In addition, the immobilization of the AgNPs and growth-factor-encapsulated nanoparticles on the GO–chitosan nanocomposites greatly enhances the antibacterial activity and osteo-inductivity, respectively [336]. Taken together, GO–polymer nanocomposites have great potential for use as multifunctional nanocomposite materials in various antibacterial applications. This is due to the substantial property enhancements and their ability to interact with various metal or metal oxide nanoparticles. Díez-Pascual and Luceño-Sánchez described several antibacterial applications of GO–polymer nanocomposites [325].

#### 4.4.1. Development of Synthetic Approaches for the Production and Use of Graphene Oxide–Polymer Nanocomposites

Various synthesis routes can be applied to prepare graphene oxide–polymer nanocomposite with covalent or non-covalent interactions [337]. Shahryari et al. summarized a range of synthesis routes in a thorough review [338]. Generally, three common methods are utilized: solution blending, melt blending, and in situ polymerization [338,339]. The solution blending approach is the most commonly used method to synthesize graphene oxide–polymer nanocomposites in a dispersion form, due to its convenience and ease of implementation. With this method, facile synthesis is achieved by simply blending the polymer with GO in solution, followed by sonication, or magnetic stirring, or shear mixing to obtain homogenous dispersions of the nanocomposite products as depicted in Figure 10 [340,341]. In Rusakova et al.’s study, an ultrasonic bath was used to disperse GO in a solution of styrene or a polyester resin in styrene, adding toluene and benzoyl peroxide and then, the mixture was placed into a special mold for polymerizing the nanocomposite [342]. 

Melt blending is applied commonly in industry due to its low cost and scalability. Unlike solution blending, melt blending eliminates the use of any toxic solvents for the dispersion. Following melting the polymer at high temperature, the GO nanofillers are dispersed into a polymer matrix by mechanical shear forces. However, it is likely to induce particle aggregation by thermal heating and local mechanical stresses during melt blending. A study employed wet phase inversion to prepare a sponge-like structure of a polymer/GO/solvent mixture as an exfoliating agent. It was subsequently ground into powder form and mixed with a polymer using melt blending [343]. The hybrid method was exemplified with two polymers, polyamide 6 (PA6) and poly (ethylene-co-vinyl acetate) (EVA). The produced nanocomposites exhibited enhanced dispersions with improved mechanical and dynamic–mechanical properties, compared with the nanocomposites prepared via melt blending [343]. 

In situ polymerization can lead to a more homogeneous dispersion of the graphene derivatives within the polymer matrix than the two methods mentioned above. In this technique, GO was initially mixed with the monomers, followed by polymerization. The polymerization can be initiated by heat or radiation. Microwave heating has been demonstrated to be a particularly efficient method to produce well-dispersed rGO polymer nanocomposites [344,345]. Hou et al. exploited microwave heating to simultaneously reduce GO and conducted a nitroxide-mediated polymerization of styrene, forming rGO-polystyrene nanocomposites [346]. 

The resulting nanocomposite could be further modified into various forms and shapes for different antibacterial applications [336,347,348].

#### 4.4.2. Application of Graphene Oxide–Polymer-Based Metal Nanocomposites in Wound Healing

Graphene oxide–polymer-based metal nanocomposites have a potential emerging role in wound healing application as in situ-forming hydrogels. Yan et al. have synthesized a novel GO–polymer-based metal nanocomposite (PEP-Ag@GO) for a wound healing application [349]. PEP-Ag@GO comprises a poly(Nisopropylacrylamide_166_-co-n-butyl acrylate_9_)-poly (ethyleneglycol)-poly(N-isopropylacrylamide_166_-co-n-butyl acrylate_9_) copolymer, denoted as PEP and AgNPs decorated on reduced GO nanosheets, denoted as Ag@GO. An aqueous mixture of the PEP-Ag@GO could transit to a hydrogel immediately in situ upon contact with the skin area that has a higher temperature than the transition temperature of PEP-Ag@GO (30 °C). The in situ formation of the hydrogel allows the treatment of wound areas that are difficult to access and minimizes tissue damage that is associated with changing of the wound dressing material [350]. More importantly, the strong interactions and bonding that arises from the PEP-GO give rise to good stability of the composite network. Therefore, the PEP-Ag@GO hydrogel resisted a transition back to liquid form at lower temperatures, even at 5 °C. This is particularly advantageous as commonly synthesized thermo-responsive hydrogels would phase transition back to a liquid at lower temperatures [351]. In vitro and in vivo experiments have also demonstrated good biocompatibility and enhanced antibacterial activity of a PEP-Ag@GO against MRSA (Mu50), leading to a much faster wound healing rate of an MRSA-infected skin defect [349]. 

#### 4.4.3. Application of Graphene Oxide–Polymer-Based Nanocomposites in Water Treatment

Membrane technologies have been widely applied in water treatments, but the growth of biofilms causes deterioration of water filtration membranes. Therefore, GO–polymer-based nanocomposites are promising materials for use in water treatments due to their antibacterial and antifouling properties. A recent study discussed newly developed GO-based nanocomposites in water treatments and identified limitations for future improvements [352]. Zeng et al. modified poly(vinylidene fluoride) (PVDF) membranes by covalently immobilizing graphene oxide quantum dots (GOQDs) to exert the antibiofouling and antibacterial properties and maintain excellent permeation properties of PVDF membranes [353]. It was found that the GOQDs–PVDF membrane inhibited the growth of *E. coli* and *S. aureus* more effectively than two-dimensional GO sheets. Cheng et al. evaluated the performances of GO-coated and GO-blended polysulfone ultrafiltration membranes and GO-coated membranes presented lower declines in water flux and higher flux recoveries than GO-blended membranes [354]. They showed strong antibacterial activity and biofouling resistance against *E. coli*. To enhance the antibacterial activity, metal nanoparticles can also be incorporated into GO–polymer-based nanocomposites. Mahmoudi et al. incorporated Ag-decorated GO nanoplates into polysulfone membranes, which demonstrated superior antibacterial properties and inhibited the formation of biofouling [355]. 

#### 4.4.4. Application of Graphene Oxide–Polymer-Based Nanocomposites in Food Packaging

Graphene oxide–polymer-based membranes are attractive candidates to be applied in food packaging due to the fact that their incorporation increases the permeability, selectivity, barrier, and antibacterial activities of packaging and prolongs the durability of the material. For example, GO was cross-linked with chitosan at 120 °C to form nanocomposite films which improved the tensile strength and thermal stability and exhibited antimicrobial properties against *E. coli* and *B. subtilis* [356]. As a result of these enhancements, the films are suitable for use in food packaging. Multiple reviews have been conducted on the development and applications of graphene oxide nanocomposites in food packaging [357,358,359,360]. However, the current state of GO–polymer composite membranes in food packaging applications are yet to be commercialized to the best of our knowledge. 

### 4.5. Lipid Polymer Hybrid Nanoparticles

Lipid polymer hybrid nanoparticles (LPHNPs) have emerged as a potentially superior nanocomposite delivery system by combining the advantages and mitigating the limitations associated with liposomes and polymeric nanoparticles alone [361,362,363,364,365,366]. LPHNPs comprise a biodegradable polymeric core for drug encapsulation, an inner lipid layer surrounding the polymeric core, and an outer polymeric stealth layer (Figure 11). The polymeric core provides high structural integrity, mechanical stability, a narrow size distribution, and higher lipophilic drug loading capacities of the LPHNPs [367]. The inner lipid layer confers the biocompatibility and delays the polymeric degradation of LPHNPs by limiting inward water diffusion, contributing to the sustained release of the composite system [368]. Finally, the outer polymeric stealth layer provides steric stabilization of the nanoparticles, acting as a stealth coating that enhances the in vivo circulation time and protect the composite system from immune recognition [369].

#### 4.5.1. Development of Lipid Polymer Hybrid Nanoparticles Using a Quality-by-Design Approach

LPHNPs are still in their early developmental stage for antibacterial applications despite the advances witnessed in cancer therapeutics [370]. The synthesis conditions of antibiotic loaded LPHNPs with desired particle characteristics is highly challenging due to the strong interplay between process variables. Therefore, a quality-by-design approach is often utilized to customize the LPHNPs in meeting the requirements or desired particle characteristics [371,372,373,374,375]. A quality-by-design approach is a statistical method that applies multiple factorial concepts and modelling to determine the interactions between two or more process variables and the desired and observed response conditions [374,376,377,378]. For instance, Dave et al. developed and optimized norfloxacin loaded LPHNPs for a topical drug delivery application [372]. They utilized a Box–Behnken design to determine the effect of process conditions including the concentration of soya lecithin (lipid) and the concentration of a polylactic acid (polymer) on the response conditions. This included parameters such as the entrapment efficiency, particle size, and cumulative drug release. It was found that the optimal norfloxacin-loaded LPHNPs have a high drug encapsulation efficiency, desired particle size with a narrow distribution range, and an improved drug release profile and stability [372]. In a study aimed at optimizing LPHNPs, Thanki et al. customized LPHNPs composed of lipidoid 5 (cationic lipid-like molecule) and poly(DL-lactic-co-glycolic acid) (PLGA) for loading antisense oligonucleotide-mediated luciferase gene (Luc-ASO) transcripts and they achieved efficient cellular delivery by using a quality-by-design approach [374]. In their study, they determined the effect of process conditions including the lipidoid 5 content and the lipidoid 5: Luc-ASO ratio against the response conditions (intensity-weighted average hydrodynamic diameter, polydispersity index, zeta potential, Luc-ASO encapsulation efficiency, Luc-ASO loading, in vitro splice-correction efficiency, and in vitro cell viability) to achieve efficient cell delivery [374]. A recent study by Ma et al. involved efficient delivery of hydroxycamptothecin (HCPT) via PEGylated LPHNPs [375]. A quality-by-design strategy was used to optimize HCPT-loaded LPHNPs with desired properties, among them, the factors representing key process conditions were the lipid/polymer mass ratio, polymer concentration, medium chain triglyceride volume, water-solvent ratio, and poly(D,L-lactide-co-glycolide) molecular weight, and the response conditions are particle size, particle size distribution, and drug-loading capacity [375]. The experimental results showed that the optimal LPHNPs had greater controlled release behavior and good stability in plasma, and effectively increased the loading of HCPT [375].

#### 4.5.2. Potential of Lipid Polymer Hybrid Nanoparticles as Antibacterial Delivery Vehicles

Recently, LPHNPs have received great attention in antibacterial applications as efficient drug delivery systems due to their combined advantages of liposomes and polymer nanoparticles [361,363,364,365,366,379]. Cai et al. have investigated the application of LPHNPs to act as promising antibacterial delivery vehicles for biofilm eradication [379]. In this study, the lipid layer was designed to contain mixed lipids of phospholipids and rhamnolipids, which acted as anti-adhesive and disrupting agents against the biofilms. The inner polymer core comprises multidrug regimens including antibiotic amoxicillin to exert antibacterial activity and the amoxicillin potentiator pectin sulfate that prevent the re-adherence of *H. pylori*. As a result, a complete eradication of *H.*
*pylori* biofilm with impaired antibacterial resistance was observed under in vitro conditions. The performance of vancomycin-loaded LPHNPs was enhanced via the synthesis using multiple lipid excipients, including glyceryl triplamitate, oleic acid, polymer excipients Eudragit RS100, chitosan, and sodium alginate [361]. Compared to LPHNPs using mono-constituents of the lipid and polymer, the LPHNPs with multiple co-excipients demonstrated higher drug loading capacities and enhanced antibacterial activity against both sensitive strains of *S. aureus* and MRSA [361]. In one recent experiment, Jaglal et al. designed a pH-responsive LPHNP system for co-delivery of vancomycin and 18β-glycyrrhetinic acid, showing its ability to eliminate 75% of MRSA in less than 12 h with the advantage of sustained and rapid release of vancomycin in acidic conditions [363]. LPHNPs consisting of a poly(lactic-co-glycolic acid) core and a dioleoyl-3-trimethylpropane lipid shell were developed for loading vancomycin and were shown to have prominent antibacterial effect against planktonic *S. aureus* cells [364]. In this study, enhanced interactions with bacterial cells and penetration into biofilms was due to the presence of lipid shells. These studies indicate that LPHNPs could be a useful strategy to deliver and enhance the antibacterial activity of the loaded drugs against planktonic cells and biofilms of diverse species. However, more studies are needed to accelerate the clinical translation of the LPHNPs in antibacterial application. 

## 5. Conclusion, Bottlenecks, and Future Perspective of Nanotechnologies Being Developed for Antibacterial Applications

This review mainly focuses on the progress and development of the prospects of nanomaterials in antibacterial applications. As described, the use of nanomaterials to combat bacterial infections has great potential for human health and medical development. Over the past few decades, significant progress has been made in understanding the antibacterial activity and potential of different classes of nanomaterials as drug carriers, leading to the discovery of a number of particularly promising candidate nanoparticle systems. Moreover, Chieruzzi et al. discussed that nanomodification such as incorporating fluoroapatite nanobioceramics into traditional clinical treatment materials, such as dental restorative glass ionomer cement, can lead to significant changes in the mechanical properties of materials, which is a very noteworthy direction for the development of new nano antibacterial materials [380]. Given their vast therapeutic potential and wide range of antibacterial applications of these nanomaterials and nanocomposites, we anticipate that more studies with emphasis on aiding the clinical translation and subsequent clinical trials of these nanotechnology-associated products will increase rapidly in the next decade. Several challenges remain to be addressed. Despite the importance of these nanomaterials as therapeutics and use in the field of biomedicine, their current limitations on human health cannot be ignored. The high-efficiency properties of nanomaterials as antibacterial agents or drug delivery vehicles allow their diffusion in different bodily organs, and sometimes the accumulation of these nanomaterials in various cells and tissues may cause negative health effects. It should be noted that the establishment of the consensus of nanomaterials physiochemical properties leading to maximum antibacterial activity and minimum toxicity is of utmost importance. With the understanding of the structure–property–activity relationship, researchers are able to reduce off-target effects of nanomaterials and effectively deliver nanotherapeutics to a desired infected tissue. For example, the use of single nanoparticles such as metal particles have many drawbacks due to their inherent toxicity. Therefore, combining a variety of nanomaterials to develop a new type of nanocomposite or nanohybrid material to obtain enhanced antibacterial activity has become a major research trend. In addition, exploring the mechanisms behind the antibacterial and physical properties of nanomaterials and nanocomposites should not be neglected. The more we learn, the better we as a community can devise new strategies to combat antimicrobial resistance. More studies in assessing the dose calibration and identification of the appropriate routes of administration for a wide range of nanomaterials are also needed. This will greatly speed up the progress towards clinical trial progression and commercialization of nanotechnology-associated end products.

## Figures and Tables

**Figure 1 nanomaterials-12-03855-f001:**
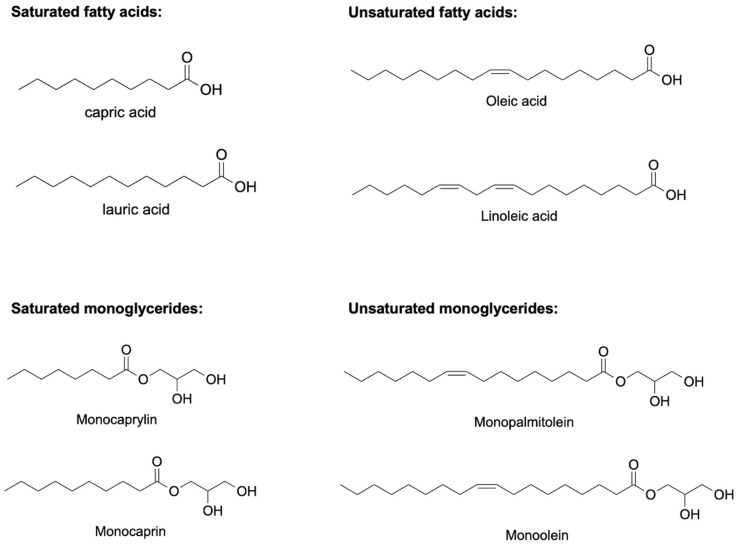
Chemical structures of some potentially antimicrobial fatty acids and monoglycerides. Saturated fatty acids include capric acid and lauric acid, unsaturated fatty acids include oleic acid and elaidic acid, saturated monoglycerides include monocaprylin and monocaprin, and unsaturated monoglycerides include monopalmitolein and monoolein.

**Figure 2 nanomaterials-12-03855-f002:**
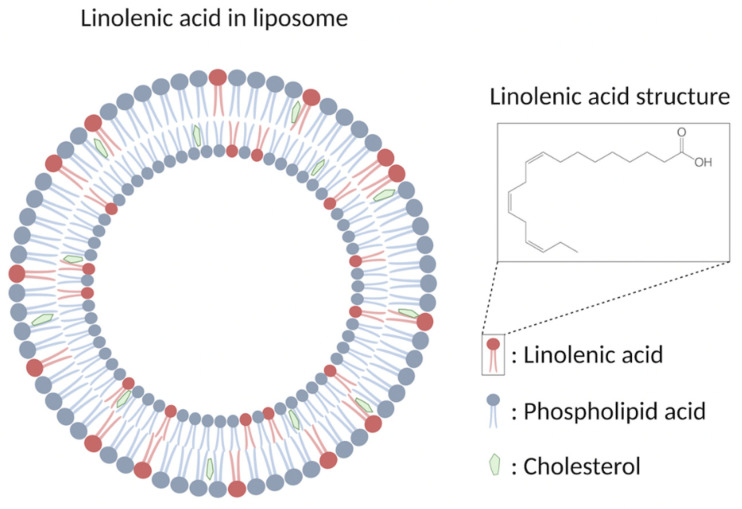
Schematic drawing showing the structure of LLA.

**Figure 3 nanomaterials-12-03855-f003:**
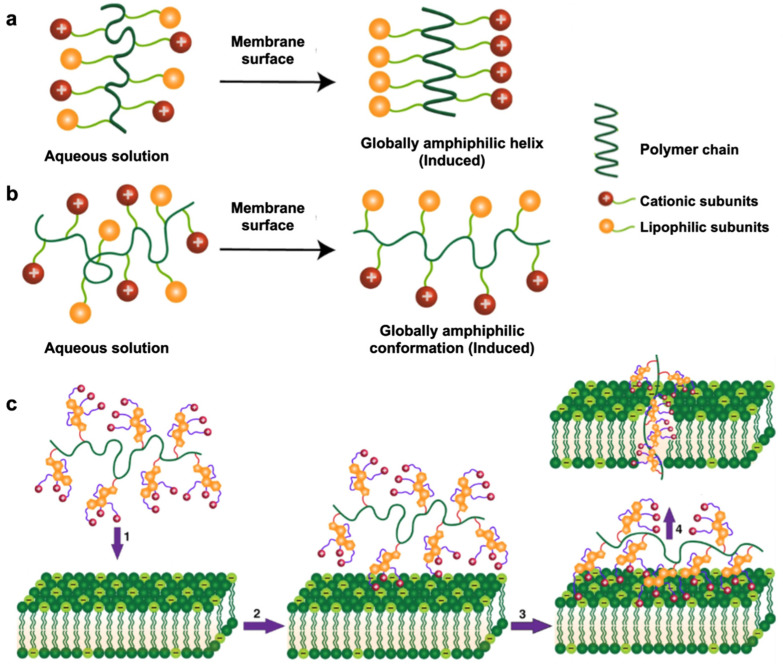
Modes of action upon contact with bacterial membrane surfaces. (**a**) Global amphiphilic helical conformation adopted by host-defense peptides; (**b**) global amphiphilic random conformation adopted by synthetic antimicrobial polymers. (**c**) Proposed antibacterial mechanism of synthetic antimicrobial polymers: (1) diffusion, (2) surface binding via cationic subunits, (3) membrane insertion via lipophilic subunits and (4) membrane disruption. (Adapted from Rahman et al., [104] 2018 with modifications.)

**Figure 4 nanomaterials-12-03855-f004:**
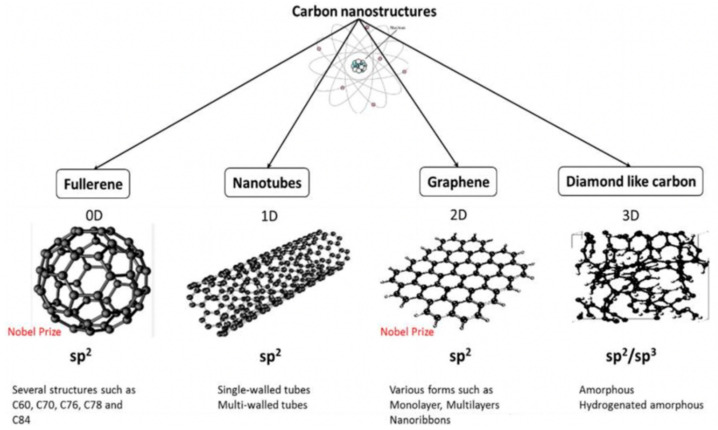
Carbon-based nanomaterials categorized via different dimensionalities (D). (Adapted with permission from Ref. [116]. 2017, Al-Jumaili et al. More details on “Copyright and Licensing” are available via the following link: https://www.mdpi.com/ethics#10 (accessed on 10 October 2022).)

**Figure 5 nanomaterials-12-03855-f005:**
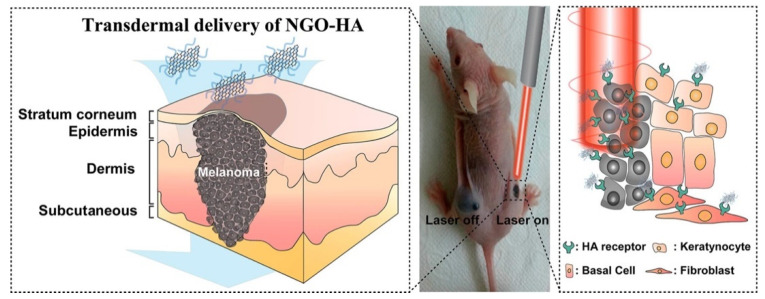
Schematic illustration for the transdermal delivery of nanographene oxide–hyaluronic acid (NGO–HA) conjugates into melanoma skin cancer cells and the following photothermal ablation therapy using a near-infrared laser. (Adapted with permission from Ref. [131]. 2014, Jung et al. More details on “Copyright and Licensing” are available via the following link: https://www.mdpi.com/ethics#10 (accessed on 10 October 2022).)

**Figure 6 nanomaterials-12-03855-f006:**
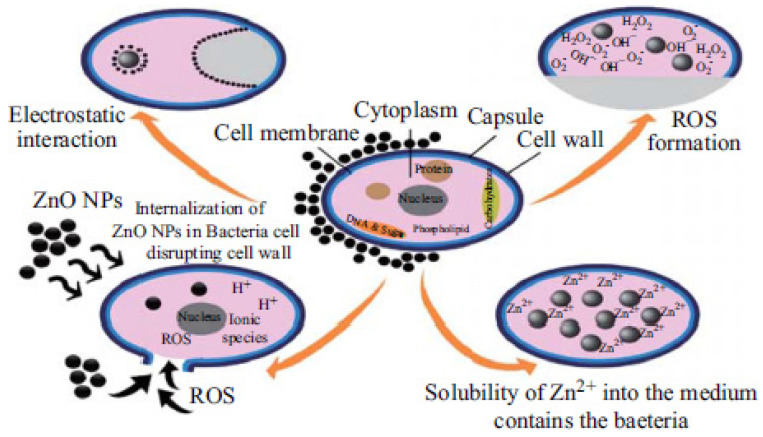
Multiple antibacterial mechanisms that are associated with ZnO nanoparticles. (Adapted with permission from Ref. [222]. 2015, Sirelkhatim et al. More details on “Copyright and Licensing” are available via the following link: https://www.mdpi.com/ethics#10 (accessed on 10 October 2022).)

**Figure 7 nanomaterials-12-03855-f007:**
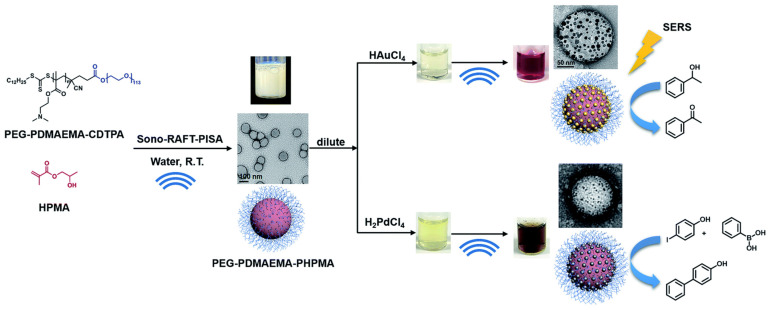
Synthesis of tertiary amine-containing polymeric nanoparticles, and in situ formation of the Au and Pd nanocomposite by ultrasound. (Adapted with permission from Ref. [233]. 2021, Wan et al. More details on “Copyright and Licensing” are available via the following link: https://www.mdpi.com/ethics#10 (accessed on 10 October 2022).)

**Figure 8 nanomaterials-12-03855-f008:**
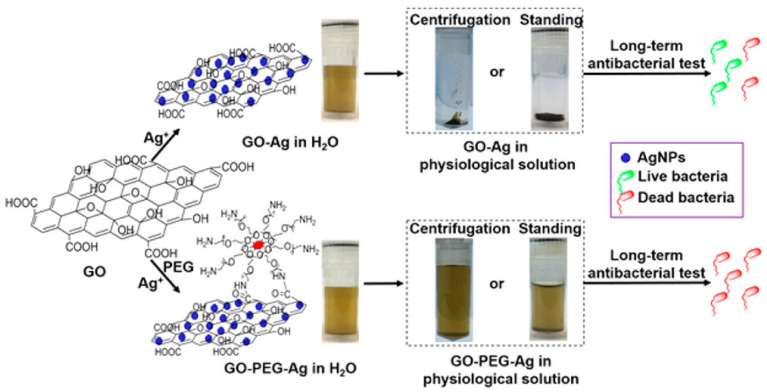
Long-term stability and antibacterial effectiveness of a PEGylated GO–Ag nanocomposite. (Adapted with permission from Ref. [309]. 2017, Zhao et al. More details on “Copyright and Licensing” are available via the following link: https://www.mdpi.com/ethics#10 (accessed on 10 October 2022).)

**Figure 9 nanomaterials-12-03855-f009:**
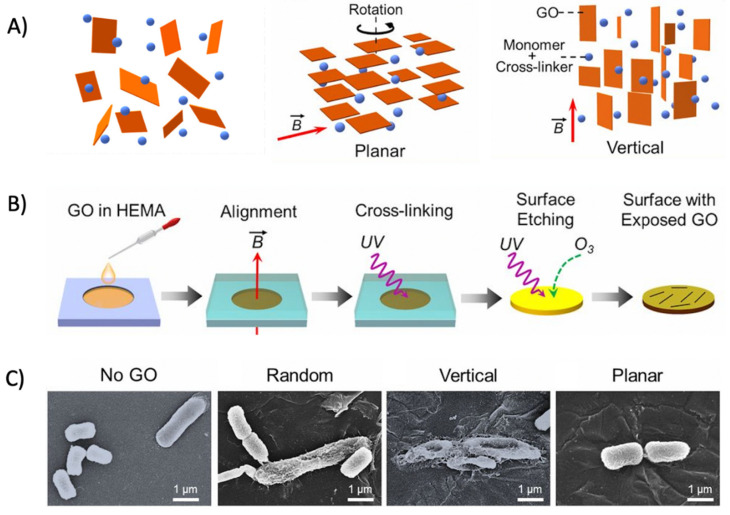
(**A**) Schematic illustration of the GO film with different orientations. From left to right: random, planar, and vertical orientations. (**B**) Schematic illustration of the film fabrication procedure. A magnetic field is applied to control the orientation of dispersed GO nanosheets, with the orientation preserved by photo-cross-linking the dispersing agents. (**C**) SEM micrographs pictured the intact *E. coli* morphology treated by no-GO film, retained *E. coli* morphological integrity treated by randomly aligned- or planar aligned-GO film and flattened and wrinkled *E. coli* morphologies after being treated by a vertically aligned-GO film. (Adapted with permission from Ref. [335]. 2017, Lu et al. More details on “Copyright and Licensing” are available via the following link: https://www.mdpi.com/ethics#10 (accessed on 10 October 2022)).

**Figure 10 nanomaterials-12-03855-f010:**
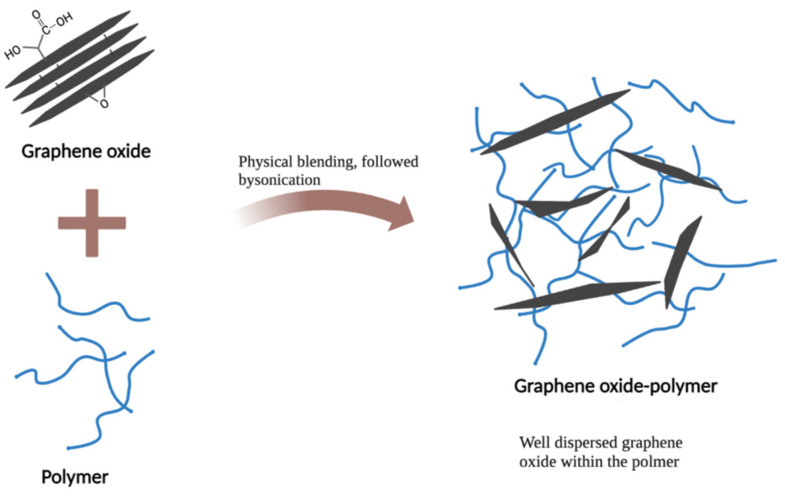
Formation of a well-dispersed graphene oxide–polymer nanocomposite with graphene being embedded in polymeric matrix.

**Figure 11 nanomaterials-12-03855-f011:**
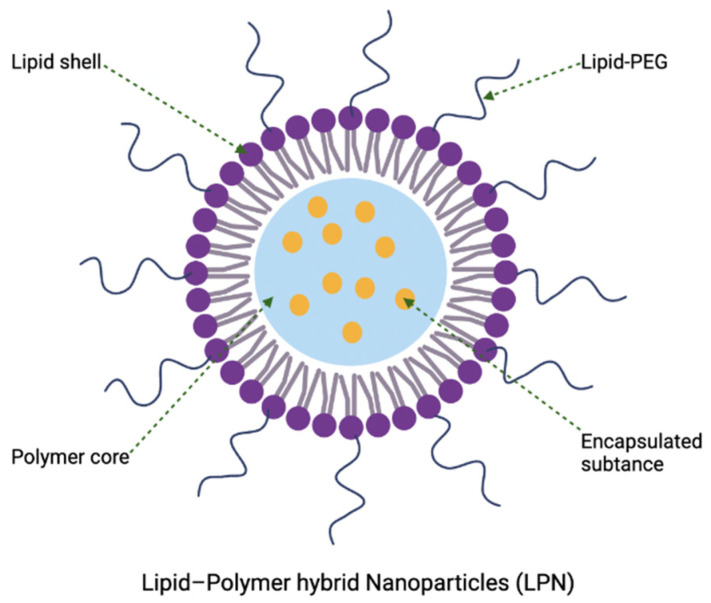
Structure of a lipid–polymer hybrid nanoparticle that comprises a polymeric core, an inner lipid layer, and an outer lipid–polyethylene glycol (PEG) shell (polymeric stealth layer).

**Table 1 nanomaterials-12-03855-t001:** Summary table of the nanomaterials.

Nanomaterials	Classes	Advantages	Disadvantages	References
Lipid	Organic	Dual functional role as antibacterial agent and nanocarrier Ease of industrial manufacturing for commercialization Good biocompatibilities	Poor colloidal stability for long-term storage Relatively weaker antibacterial activity	[16,17]
Polymer	Organic	Dual functional role as antibacterial agent and nanocarrier Strong bactericidal activity Good colloidal integrity and stability	Poor biocompatibilities	[18,19]
Carbon	Organic	Dual functional role as antibacterial agent and nanocarrier Highest drug loading capacity Strong bactericidal activity with physical and chemical antibacterial mechanism	Higher tendency of agglomeration Low water solubility	[20,21,22]
Metal	Inorganic	Strong bactericidal activity Multiple antibacterial applications for dry (coating) and wet environment (disinfectant) Ease of industrial manufacturing for commercialization	Higher tendency of agglomeration Poor biocompatibilities Lack of delivery ability	[23,24,25]
Metal oxide	Inorganic	Good biocompatibilities Photosensitizing agents with multiple antibacterial mechanisms Ease of industrial manufacturing for commercialization	Higher tendency of agglomeration Lack of delivery ability Environmental hazards especially to aquatic environment	[26,27]

**Table 2 nanomaterials-12-03855-t002:** Liposomal nanoformulation in clinical development for antibacterial therapy.

Product Name	Encapsulating Materials	ClinicalTrials.gov Identifier	Description
-	AP10-602/ GLA-SE	NCT02508376	Trial on the safety, tolerability, and immunogenicity of the vaccine candidates for the protection against tuberculosis
-	CAL02	NCT02583373	Trial on broad-spectrum antitoxin agent CAL02 that neutralizes bacterial toxins to protect against infection severity and deadly complications
Pulmaquin	Ciprofloxacin	NCT02104245	Trial on Pulmaquin^®^ in the management of chronic lung infections in patients with non-cystic fibrosis bronchiectasis
MAT2501	Amikacin	-	Orally administered amikacin liposomal formulation for various MDR infections that completed Phase 1 study
CAF01	Tuberculosis Subunit Vaccine Ag85B-ESAT-6	NCT00922363	Trial on the safety of new liposomal vaccine adjuvant for protection against tuberculosis

**Table 3 nanomaterials-12-03855-t003:** Antibacterial activity of metal nanoparticles against different bacteria.

NPs	Target Bacteria	References
Ag	*Acinetobacter baumannii*, *Salmonella typhi*, *Vibrio cholerae, Bacillus subtilis, S. aureus*, MDR *E. coli*, *Streptococcus pyogenes*, *P. aeruginosa*, coagulase-negative *S. epidermis*, *E. faecalis*, *K. pneumoniae*, *Listeria monocytogenes*, *Proteus mirabilis*, *Micrococcus luteus*	[23,190,191,192,193]
Au	*E. coli*, *S. aureus*, *B. subtilis*, *K. pneumoniae*, *S. epidermidis*, *P. aeruginosa, L. monocytogenes, Salmonella typhimurium*	[194,195,196,197,198,199]
Cu	*Enterobacter aerogenes*, *E. coli*, *Klebsiella oxytoca*, *S. aureus*, *S. pyogenes, B. subtilis*	[200,201,202,203,204]
Bi	*Streptococcus mutans*, *C. albicans*, *E. faecalis*	[205,206,207,208]
Cu/Zn bimetal NPs	*E. coli*, *S. aureus*, MRSA, *Alcaligenes faecalis*, *Citrobacter freundii*, *K. pneumoniae*, *Clostridium perfringens*	[209,210,211]
Ag/Cu bimetal NPs	*E. coli, S. aureus, A. faecalis*, *C. freundii*, *K. pneumoniae*, *C. perfringens*, *P. aeruginosa, B. subtilis*	[211,212,213]
Superparamagnetic iron oxide NPs coated with Ag or Au	*E. coli*, *S. aureus*, *P. aeruginosa*, *E. faecalis*, *S. epidermidis*	[214]

Abbreviations: NPs, nanoparticles; MDR, multidrug resistant; MRSA, methicillin-resistant *Staphylococcus aureus*.

**Table 4 nanomaterials-12-03855-t004:** Versatility of polymer-matrix metal nanocomposites for various antibacterial applications.

Structures/Forms	Potential Antibacterial Applications	References
Film	Surface coating Food packaging Wound dressing	[252] [253] [254]
Scaffold	Bone tissue engineering Wound dressing	[255] [256]
Membrane	Wastewater treatment/water filtration	[257]
Sponge	Wound dressing	[258]
Gel	Antifouling/surface coating Tissue engineering Wound healing	[259] [260] [261]

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
