# Peer review of "Recent Advances in the Development of Lipid-, Metal-, Carbon-, and Polymer-Based Nanomaterials for Antibacterial Applications"

_nanomaterials, 2022, doi:10.3390/nano12213855_

Round 1

Reviewer 1 Report

Very interesting narrative review of the literature on research in terms of nanomaterials. Work well done and smooth. Some criticisms are however present:

Check that all keywords are pubmed mesh terms

- Insert in the title the reference to the narrative review of the literature

-In the Introduction section, some considerations should be added on biomaterials which, in addition to nanotechnologies, are gaining interest in the international scientific literature, as frontiers in research. In this regard, I recommend that you insert the following scientific work in the reference section, which could be of help to the reader:

Lardani L, Derchi G, Marchio V, Carli E. One-Year Clinical Performance of Activa ™ Bioactive-Restorative Composite in Primary Molars. Children (Basel). 2022; 9 (3): 433. Published 2022 Mar 19. doi: 10.3390 / children9030433

- Even if it is a narrative review of the literature at the end of the introduction section some general information on the research strategies adopted should be added (inclusion criteria, year, etc.)

-In order to simplify the reader's understanding, I would recommend inserting a summary table of the different nanomaterials at the beginning of the discussion

-The images of the figures are very blurry; increase the resolution

-Check the syntax of the English language by a native speaker.

-At the end of each paragraph the advantages and disadvantages of the single material I would highlight it in order to improve understanding.

In the discussion section, another aspect to emphasize is the research attempts to enrich traditional materials with antimicrobial agents. In this regard, I recommend that you insert the following scientific work in the reference section, which could be of help to the reader:

Chieruzzi, M .; Pagano, S .; Lombardo, G .; Marinucci, L .; Kenny, J.M .; Torre, L .; Cianetti, S. Effect of Nanohydroxyapatite, Antibiotic, and Mucosal Defensive Agent on the Mechanical and Thermal Properties of Glass Ionomer Cements for Special Needs Patients. J. Mater. Res. 2018, 33, 638–649

Reviewer 2 Report

The manuscript entitled “Recent Advances in The Development of Nanomaterials for Antibacterial Applications” described the recent advancements in the design and development of nano and nano-composite materials to fight multi-drug resistant bacteria. However, the manuscript suffers few shortcomings as described below: 

1.     It is not very clear why the authors are specifically focusing on some specific materials like graphene. It should be emphasized. Based on that the title should be changes. General term like nanomaterials is misleading in this paper.

2.     Also, it would be better to include one paragraph about other inorganic materials with antibacterial properties. Authors have mentioned some materials in section 3.1, but that should be supported with lot of references for each material (at least 4 or 5 references for each material, since it is a review article).

3.     I feel it is better to add fluoride in this list. It is one of materials to treat oral bacteria for long time. Some references are given here. Authors also check for other references either old or recent (eg, Journal of Advanced Research Vol. 8, Issue 4, (2017) 387-392; Iranian Journal of Pediatrics: Vol.31, issue 3; e111422; Acta Biomaterialia 79 (2018) 148-157; BMC Oral Health volume 21, Article number: 175 (2021))

4.     The authors have mentioned “look forward to further development of these materials” in the abstract. This line can be altered as “look forward to further develop these materials”. (Page 1 -Line 40)

5.     The authors have mentioned “In short, it these be classified into lipidic nanocarriers and lipidic nanoparticles.” in Section 2.1. This line can be altered as “In short these can be classified into lipidic nanocarriers and lipidic nanoparticles.”. (Page 3 -Line 111)

6.     What is the difference between physiological or physiological-related lipids as the authors have mentioned in Section 2.1.1. (Page 3 -Line 118)

7.     The number of atoms in the molecular formula of magnetite and copper oxide should be given in the subscript. (Page 4 -Line 154)

8.     The authors can explain in short what polytherapy is for better understanding of the readers (Page 4 -Line 161)

9.     The authors should replace “so” in place of “thus” (Page 5 -Line 204)

1.  The authors have mentioned about linoleic acid when it should be linolenic acid. (Page 6-Line 216)

1.  The authors have mentioned about second generation of lipid nanoparticles. Give an example how they have been employed for better antibacterial activity (Page 6- Line 233)

1.  The authors have mentioned in Figure 4 that quantum dots are 2D materials. But they are 0D materials. (Page 10)

1.  The text in Figure 6 should be clearer and more distinct for the ease of the readers. (Page 14)

1.  The first two lines in section 4.1.3 have to be re-written for better understanding (Page 16 -Line 627)

1.  The authors have mentioned anbacterial effects instead of antibacterial effects. (Page 16 -Line 638)

1.  The authors have mentioned about doped nano clay under section 4.2.1. What is the purpose of it and does it have any influence on the antibacterial properties? (Page 18 -Line 696,697,698).

1.  What is Lipidoid 5 mentioned under the section 4.5.1 (Page 27 -Line 1046)

1.  The authors have to alter the sentence mentioned for better understanding – (Page 28- Line 1109)

Reviewer 3 Report

See the attachment please

Round 2

Reviewer 2 Report

Authors are addressed almost all the comments given to them. Some references given by the reviewers was not included in the revised manuscript. Reviewer 3 point 3 (Acta Biomaterialia 79 (2018) 148-157). 

If any issues with any comments by the reviewers, it should be justified.

Further, I am concerned about the the quality of the figures. All are looking blurred. If the figures are taken from other publications, high resolution images should be downloaded and incorporated properly. Authors should work on it. 

Author Response

Authors are addressed almost all the comments given to them. Some references given by the reviewers was not included in the revised manuscript. Reviewer 3 point 3 (Acta Biomaterialia 79 (2018) 148-157). 

If any issues with any comments by the reviewers, it should be justified.

Thanks for pointing this out. We have added the suggested reference to this review paper (ref 161)

Further, I am concerned about the quality of the figures. All are looking blurred. If the figures are taken from other publications, high resolution images should be downloaded and incorporated properly. Authors should work on it. 

Thank you for pointing this out. All Figures was downloaded through the high-resolution version from original publications. However, the manuscript sent to reviewers seems to be compressed online. We have contacted the editorial team about this issue and those high-resolution images were emailed to the editor separately.